
# Yangian algebra and correlation functions in planar Gauge theories

**Niklas Beisert[1,2]⋆ and Aleksander Garus[1]†**

**1** Institut für Theoretische Physik, Eidgenössische Technische Hochschule Zürich,
Wolfgang-Pauli-Strasse 27, 8093 Zürich, Switzerland
**2** Kavli Institute for Theoretical Physics,
University of California, Santa Barbara, CA 93106-4030, USA

⋆ nbeisert@itp.phys.ethz.ch
† agarus@itp.phys.ethz.ch

## Abstract

In this work we consider colour-ordered correlation functions of the fields in integrable planar gauge theories such as $\mathcal{N} = 4$ supersymmetric Yang–Mills theory with the aim to establish Ward–Takahashi identities corresponding to Yangian symmetries. To this end, we discuss the Yangian algebra relations and discover a novel set of bi-local gauge symmetries for planar gauge theories. We fix the gauge, introduce local and bi-local BRST symmetries and propose Slavnov–Taylor identities corresponding to the various bi-local symmetries. We then verify the validity of these identities for several correlation functions at tree and loop level. Finally, we comment on the possibility of quantum anomalies for Yangian symmetry.

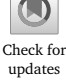
# 1 Introduction

Integrability is a tremendously useful feature of selected theoretical physics models that makes these models much more amenable to exact calculations. At a technical level, integrability is often viewed as a (more or less hidden) enhancement of the model's symmetries, and the extended symmetries eventually explain the efficiency of the integrability-based methods. An exciting class of integrable models of recent interest consists of certain quantum gauge field theories featuring prominently in the AdS/CFT correspondence, namely $\mathcal{N} = 4$ supersymmetric Yang–Mills (sYM) theory and several other related models. Investigating and exploiting the integrable structures in these models has led to many novel insights about them and about the AdS/CFT correspondence in the past 15 years; see [1] for a review. The relationship between integrability of planar gauge theories and the enhancement of the spacetime symmetries to a Yangian algebra has been known for a long time [2], see also [3–7]. Only much more recently, it has been shown that the model's action is in fact perfectly invariant under this Yangian symmetry, at least classically [8,9].

Within field theories, symmetries typically imply the existence of conserved currents by means of Noether's theorem. In this case, however, the extra symmetries are in some sense non-local whereas Noether's theorem assumes a local symmetry action. Nonetheless Yangian

symmetry has been demonstrated to lead to non-trivial relationships for several types of observables. Most prominently, so-called dual conformal symmetry [10] as a part of Yangian symmetry [11] has been essential for constraining and constructing planar colour-ordered gluon scattering amplitudes, see e.g. [12–16]. Unfortunately, the Yangian symmetries are somewhat obscured beyond tree level due to the appearance of infra-red divergences related to the presence of massless particles. Therefore, the enhanced symmetries have largely been used either at the level of loop integrands or in a modified form for finite remainder functions. In the former approach, divergences are avoided and therefore Yangian symmetry applies fully. However, the results are of questionable use because of the need to regularise before integrating. In the latter approach, the divergent part is subtracted according to some scheme, and the effect of the symmetries acting on it is summarised by some effective contribution complementing the action on the finite part. While the approach leads to useful results [17–20], the construction of the effective contributions to the symmetries is scheme-dependent and sometimes appears ad hoc.

In this work our aim is to establish an opposite approach and derive Ward–Takahashi identities for planar colour-ordered correlation functions based on the recently established Yangian symmetries of the action. Correlation functions of individual fields at distinct spacetime points are perfectly finite observables of a renormalised quantum field theory. Therefore they are also perfectly suited for symmetry considerations. Unfortunately, correlation functions of individual fields are not gauge-invariant and therefore they are not, by themselves, well-defined observables of a gauge theory. However, they do contain gauge-invariant information, and their unphysical contributions can be computed unambiguously after having fixed a particular gauge.[1] Gauge fixing typically adds extra unphysical degrees of freedom, such as ghost fields, to the system which are in no way bound to follow the representation theory of Yangian symmetry. Nevertheless, we can set up Ward–Takahashi identities which take the gauge-fixing procedure into account and which merely constrain the gauge-invariant information contained in the gauge-fixed correlation functions. Another complication is that the formulation of Yangian symmetry [8,9] is intertwined with gauge symmetry and breaking the latter has some impact on the former. We will thus have to understand aspects of the Yangian algebraic relations in detail, and we shall find that curiously the ordinary gauge symmetries are enhanced to non-local symmetries in the planar gauge theory. These symmetries as well as the corresponding BRST symmetries after gauge fixing play a crucial role in the Yangian Ward–Takahashi identities for correlation functions.

The present article is organised as follows: In the following Sec. 2 we review the formulation of bi-local symmetries for planar gauge theories put forward in [8,9]. The next Sec. 3 discusses the relations of the Yangian algebra and how they give rise to non-local enhancements of gauge symmetries. Subsequently, we fix the gauge by the Faddeev–Popov method and introduce BRST symmetry in Sec. 4. We also show how to formulate an invariance statement for the gauge-fixed action which in fact requires a non-local generalisation of BRST symmetry. In Sec. 5 we propose a set of Slavnov–Taylor identities to formulate Yangian symmetry for quantum correlators. In order to derive the latter, we introduce a notion of bi-local total variations within the planar path integral. Such a notion is by no means established, nor is it clear whether it can be defined in a meaningful way at all. Nevertheless, we arrive at a set of identities which can be formulated and tested in the ordinary framework. Verifying the identities is the subject of Sec. 6 where we apply them to planar colour-ordered correlation functions of the fields. This gives rise to Ward–Takahashi identities which we demonstrate to hold for correlators of three and four fields at tree level. Properly taking all effects of gauge fixing into account leads to a flurry of extra terms which all cancel eventually in all considered

---

[1]Evidently, the results depend on the choice of gauge, on how to represent the symmetries on the unphysical degrees of freedom, on how the latter are renormalised and so on.

cases. We delegate a full treatment of gauge-fixed correlation function to App. A. Finally in Sec. 7, we shall discuss Yangian symmetry at the loop level in order to understand whether quantum effects may break the symmetry. In particular, we will lay out an argument against the existence of a quantum anomaly of Yangian symmetry in planar $\mathcal{N} = 4$ sYM. We conclude in Sec. 8 and give an outlook on open questions.

## 2  Bi-local symmetries in planar gauge theory

We start by introducing some of the basic concepts of gauge theory and Yangian algebras which will be relevant to the present article. Subsequently, we will review the results of [8, 9] on Yangian invariance in planar gauge theory.

### 2.1  Planar gauge theory

In this work we will be concerned with gauge theory models which become integrable in the planar limit, see [1]. Along the lines of [8, 9] we will consider the feature of integrability as a manifestation of Yangian symmetry. We will base our analysis largely on the extended algebraic properties of these models proposed and discussed in [8, 9] to be reviewed in the later parts of this section. There will be no need to introduce and discuss specifics such as field content or action, we will merely make reference to elementary concepts of these models. Instead, we will discuss some relevant subtleties in more detail than usual in order to make the subsequent investigations more accessible.

The gauge theory has a number of fields which we will collectively denote by $Z^I$ with $I, J, \ldots$ some multi-index enumerating the different kinds of fields. Among these, we will only need the gauge field $A_\mu$ explicitly. For concreteness, we shall assume the gauge group to be $U(N_c)$ and all (real) fields to be $N_c \times N_c$ (hermitian) matrices as in $\mathcal{N} = 4$ sYM theory.[2] Importantly, in all of these models, the fields can be composed by the matrix product to *field monomials*

$$\mathcal{X} = Z^I Z^J Z^K \ldots. \tag{2.1}$$

The sequence of the individual matrix fields matters, and at sufficiently large $N_c$ there are no identities to relate different orderings. This means that in the planar limit $N_c \to \infty$ all monomials are independent and form a basis for polynomials. The above *open monomial* $\mathcal{X}$ is an $N_c \times N_c$ matrix, and therefore it inevitably transforms non-trivially under the gauge group. Singlet combinations of the fields are obtained by taking the trace over colour space

$$\mathcal{O} = \mathrm{tr}(Z^I Z^J Z^K \ldots). \tag{2.2}$$

Apart from relating the last and first fields, the trace implies a cyclic relationship

$$\mathrm{tr}(Z^I Z^J Z^K \ldots Z^P Z^Q) = \mathrm{tr}(Z^J Z^K \ldots Z^P Z^Q Z^I), \tag{2.3}$$

therefore we call $\mathcal{O}$ a *cyclic monomial*. In addition to cyclic polynomials, we will also require the notion of *closed polynomials* at intermediate stages of calculations. Closed polynomials are defined analogously to open polynomials (2.1), but there is an additional implicit neighbouring relationship between the last and the first site, so that the sites form a periodic sequence. We shall view cyclic polynomials as the subspace of closed polynomials invariant under cyclic permutations, and the cyclic equivalence relation will be denoted by the symbol '$\simeq$'

$$(Z^I Z^J Z^K \ldots Z^P Z^Q) \simeq (Z^J Z^K \ldots Z^P Z^Q Z^I). \tag{2.4}$$

---

[2]Other relevant models such as ABJM theory have somewhat different gauge groups and fields, which can be viewed as restrictions of bigger matrices to some subset.

We assume that the Lagrangian $\mathscr{L}$ and the action $\mathscr{S}$ are given by cyclic polynomials[3]

$$\mathscr{S} = \int \mathrm{d}x^d\, \mathscr{L}, \qquad \mathscr{L} = \mathrm{tr}(Z^I Z^J Z^K \ldots) + \ldots . \tag{2.5}$$

In the case of cyclic polynomials integrated over spacetime, such as the action $\mathscr{S}$, the equivalence relationship '$\simeq$' is meant to also include integration by parts.

Note that for notational simplicity we consider all fields $Z^I$ to be bosonic. The discussion will equally apply in the presence of fermions when the appropriate signs due to statistics are inserted; we will only make such signs explicit when discussing particular fields, in particular the Faddeev–Popov ghost fields $C, \bar{C}$ to be introduced later.

Gauge transformations will be denoted by the generator $\mathrm{G}[\Lambda]$ where $\Lambda$ is some $N_c \times N_c$ matrix of fields. It acts on the fields of the model as

$$\mathrm{G}[\Lambda]\cdot Z := [\Lambda, Z], \qquad \mathrm{G}[\Lambda]\cdot A_\mu := i\partial_\mu \Lambda + [\Lambda, A_\mu]. \tag{2.6}$$

Consequently, the generators obey an algebra $[\mathrm{G}[\Lambda_1], \mathrm{G}[\Lambda_2]] = \mathrm{G}[\Lambda_3]$ with $\Lambda_3 = \mathrm{G}[\Lambda_1]\cdot\Lambda_2 - \mathrm{G}[\Lambda_2]\cdot\Lambda_1 - [\Lambda_1, \Lambda_2]$. The resulting expression $\Lambda_3$ can be simplified depending on the type of gauge parameter fields $\Lambda_1$ and $\Lambda_2$. In particular, for an external field $\Lambda_2$, the gauge transformation $\mathrm{G}[\Lambda_1]$ will act trivially by construction, $\mathrm{G}[\Lambda_1]\cdot\Lambda_2 = 0$. In our article, we will almost exclusively consider the gauge parameters to be internal fields of the theory or covariant combinations $\mathscr{X}$ thereof for which $\mathrm{G}[\Lambda]\cdot\mathscr{X} = [\Lambda, \mathscr{X}]$. We thus get a simplified algebra for covariant internal transformation parameters $\mathscr{X}_1$ and $\mathscr{X}_2$ which are open polynomials

$$\big[\mathrm{G}[\mathscr{X}_1], \mathrm{G}[\mathscr{X}_2]\big] = \mathrm{G}\big[[\mathscr{X}_1, \mathscr{X}_2]\big]. \tag{2.7}$$

The gauge field serves as the connection for a gauge-covariant derivative $\nabla$ which is defined on a covariant field $Z$ as

$$\nabla_\mu Z := \partial_\mu Z + i[A_\mu, Z]. \tag{2.8}$$

Acting on another gauge field, the covariant derivative shall be defined as the field strength

$$\nabla_\mu A_\nu = -\nabla_\nu A_\mu = F_{\mu\nu} := \partial_\mu A_\nu - \partial_\nu A_\mu + i[A_\mu, A_\nu]. \tag{2.9}$$

The commutator of covariant derivatives reads

$$[\nabla_\mu, \nabla_\nu]Z = i\mathrm{G}[F_{\mu\nu}]\cdot Z. \tag{2.10}$$

## 2.2 Yangian symmetry

The gauge theory model has some ordinary spacetime symmetries which form a Lie algebra $\mathfrak{g}$. The latter is spanned by the generators $\mathrm{J}^a$, $a = 1, \ldots, \dim \mathfrak{g}$, and their algebra is given by the structure constants $f$

$$[\mathrm{J}^a, \mathrm{J}^b] = if^{ab}{}_c \mathrm{J}^c. \tag{2.11}$$

For notational simplicity we shall again pretend that all generators $\mathrm{J}^a$ are bosonic; however, the discussion will equally apply to Lie superalgebras when appropriate signs due to statistics are inserted.

---

[3]It is conceivable that the discussion can be extended to field theory models with fundamental matter fields and open polynomial contributions to the Lagrangian. However, this will require substantial modifications.

For concreteness, we shall assume the underlying spacetime to be commutative and flat, so that translations will be among the spacetime symmetries.[4] Translations are represented by the momentum generator $P_\mu$, and commutativity of spacetime implies that the momentum generators commute, $[P_\mu, P_\nu] = 0$.

For integrable planar gauge theory models as discussed in [8, 9], the ordinary spacetime symmetries extend to a Yangian quantum algebra $Y[\mathfrak{g}]$. The ordinary symmetries of the algebra $\mathfrak{g}$ form the zeroth level of the Yangian algebra, and they are required to have an invertible invariant quadratic form. In addition, there is one level-one generator $\widehat{J}^a$ corresponding to each level-zero generator $J^a$. These generators transform in the adjoint representation of $\mathfrak{g}$

$$[J^a, \widehat{J}^b] = i f^{ab}{}_c \widehat{J}^c, \tag{2.12}$$

and they obey the so-called Serre relation

$$\left[\widehat{J}^a, [\widehat{J}^b, J^c]\right] + \text{cyclic} = f^{ad}{}_e f^{bf}{}_h f^{cg}{}_i f_{dfg} J^{(e} J^h J^{i)}, \tag{2.13}$$

where the indices of the last structure constant $f_{dfg}$ have been lowered by means of the inverse quadratic invariant form. Furthermore, the Yangian algebra contains infinitely many higher-level generators. However, we can ignore these because their algebraic relations are fully determined by the above relations involving only level-zero and level-one generators.

The Yangian algebra is a Hopf algebra, and it has a non-trivial coproduct which implies the following representations on a tensor product of $n$ factors

$$J^a = \sum_{j=1}^n J_j^a, \qquad \widehat{J}^a = \sum_{j=1}^n \widehat{J}_j^a + f^a{}_{bc} \sum_{j=1}^{n-1} \sum_{k=j+1}^n J_j^b J_k^c. \tag{2.14}$$

Here $J_j^a$ and $\widehat{J}_j^a$ denote the representations of $J^a$ and $\widehat{J}^a$, respectively, on site $j$ of the tensor product, and $f^a{}_{bc}$ is another version of the structure constants with one index lowered. Following the structure of the tensor product representations, the level-zero generators $J^a$ are called *local* because they act locally on the tensor product. Conversely, the level-one generators $\widehat{J}^a$ are called *bi-local* because they act on all pairs of tensor factors, even at a distance. The level-one generators additionally act by local contributions. Note that the bi-local contributions in the second term are completely determined by the level-zero generators whereas the freedom in specifying the level-one representation lies in tuning the local contributions in the first term.

One helpful tool to avoid writing out indices and structure constants is Sweedler's notation: for a level-one generator $\widehat{J} = \widehat{J}^a$, we define the bi-local combination of level-zero generators appearing in the tensor product representation (2.14) as

$$J^{(1)} \otimes J^{(2)} := f^a{}_{bc} J^b \otimes J^c. \tag{2.15}$$

It allows us to write the above tensor product representation at level one compactly as $\widehat{J} = \sum_j \widehat{J}_j + \sum_{j<k} J_j^{(1)} J_k^{(2)}$. Note that anti-symmetry of the structure constants implies anti-symmetry of Sweedler's notation

$$J^{(1)} \otimes J^{(2)} = -J^{(2)} \otimes J^{(1)}. \tag{2.16}$$

---

[4]This assumption may not be essential for the overall applicability of the article, but a concrete momentum generator will help to illustrate some features.

## 2.3   Non-linear representations

A complication in formulating Yangian symmetries for planar gauge theory models is that the action of Yangian generators should be formulated in a way that is compatible with gauge transformations. Therefore it is advisable to work with gauge-covariant representations which map all covariant fields to some covariant combination of the fields. This necessarily implies that the representation is non-linear in the fields. So-called *non-linear representations*[5] appear naturally in the context of interacting (quantum) field theory, but usually only for local generators J. The generalisation from local to bi-local generators $\widehat{J}$ has some rather non-trivial issues, some of which were addressed in [8, 9]. Here we will review non-linear local representations and present some tools to work with them. Later on we shall generalise to non-linear bi-local actions.

**Ideal of gauge transformations.**   To understand the necessity and implications of non-linear representations in a gauge theory, let us consider translations. Translations are represented by the momentum generator $P_\mu$ which acts on any field $Z$ by the covariant derivative[6]

$$P_\mu \cdot Z = i\nabla_\mu Z. \tag{2.17}$$

Due to the presence of the gauge field $A_\mu$ within the covariant derivative, this representation is clearly non-linear in the fields.

   Covariance of the representations also has implications on the algebraic structure. Due to commutativity of spacetime, one would expect the momentum generators to commute. However, commutativity is mildly violated by a gauge transformation sourced by the gauge field strength $F$

$$[P_\mu, P_\nu] = -iG[F_{\mu\nu}]. \tag{2.18}$$

Happily, the gauge transformations G form an ideal of the complete symmetry algebra according to the algebraic relation

$$\left[G[\mathscr{X}], J\right] = -G[J\cdot\mathscr{X}] \tag{2.19}$$

along with the closure (2.7) of gauge transformations. Therefore, it is possible to isolate the spacetime transformations by quotienting out gauge transformations. In other words, the algebra reduces to spacetime transformations when acting on gauge-invariant states.

**Non-linear actions.**   Let us now understand how non-linear representations act on field polynomials. To that end, it is instructive to expand an open polynomial $\mathscr{X}$ in terms of the number of fields $n$ as $\mathscr{X}_{[n]}$

$$\mathscr{X} = \sum_n \mathscr{X}_{[n]}. \tag{2.20}$$

Likewise, we shall denote the contribution to a non-linear representation J that adds $m$ fields to the object it is acting upon by $J_{[m]}$. The expansions reads

$$J = \sum_m J_{[m]}. \tag{2.21}$$

Altogether we can write the non-linear action of J on $\mathscr{X}$ as

$$J\cdot\mathscr{X} = \sum_{n,m} J_{[m]}\cdot\mathscr{X}_{[n]} = \sum_n \sum_{m=0}^n J_{[m]}\cdot\mathscr{X}_{[n-m]} = \sum_n (J\cdot\mathscr{X})_{[n]} \tag{2.22}$$

---

[5]Non-linear representations merely act non-linearly on single fields, but they act linearly on the vector space of field polynomials.

[6]The corresponding non-covariant representation $P_\mu \cdot Z = i\partial_\mu Z$ merely uses the partial derivative; consequently, it is linear in the fields, but it produces undesirable non-covariant expressions within the gauge theory context.

with the expansion coefficients

$$(\mathrm{J}\cdot\mathscr{X})_{[n]} = \sum_{m=0}^{n}\sum_{k=1}^{n-m} \mathrm{J}_{[m],k}\mathscr{X}_{[n-m]}. \tag{2.23}$$

Here $\mathrm{J}_{[m],k}\mathscr{X}_{[n]}$ is an operator which replaces a field $Z$ at site $k$ of the polynomial $\mathscr{X}_{[n]}$ by the sequence $\mathrm{J}_{[m]}\cdot Z$ of length $m+1$. All the sites $j = 1,\dots,k-1$ of the polynomial are left untouched, while the sites $j = k+1,\dots,n$ are shifted by $m$ sites to $j = k+m+1,\dots,n+m$.

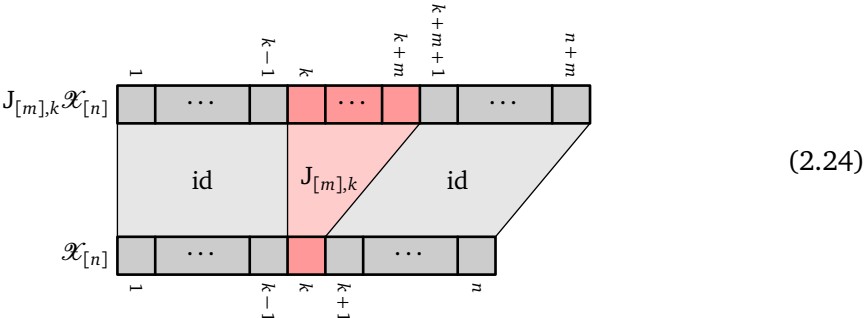

$$\tag{2.24}$$

Based on the above construction, it is straight-forward to determine the commutator of two non-linear actions $\mathrm{J}^a$ and $\mathrm{J}^b$ in the expansion

$$[\mathrm{J}^a,\mathrm{J}^b] = \sum_{n}[\mathrm{J}^a,\mathrm{J}^b]_{[n]}. \tag{2.25}$$

The expansion coefficients read

$$[\mathrm{J}^a,\mathrm{J}^b]_{[n],j} = \sum_{m=0}^{n}\sum_{k=1}^{m+1} \mathrm{J}^a_{[n-m],k+j-1}\mathrm{J}^b_{[m],j} - \sum_{m=0}^{n}\sum_{k=1}^{m+1} \mathrm{J}^b_{[n-m],k+j-1}\mathrm{J}^a_{[m],j}. \tag{2.26}$$

Note that the non-overlapping terms drop out from the commutator as usual due to the non-linear commutation relation

$$\mathrm{J}^a_{[l],j}\mathrm{J}^b_{[m],k} = \mathrm{J}^b_{[m],k+l}\mathrm{J}^a_{[l],j} \qquad \text{if } j < k. \tag{2.27}$$

In the diagrammatical notation of (2.24) this relation has the following form:

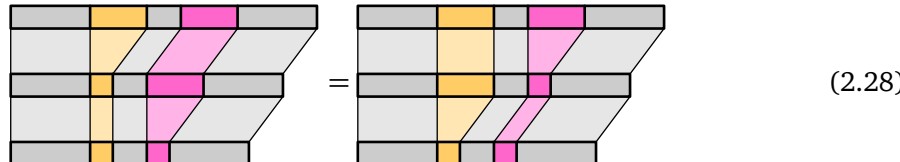

$$\tag{2.28}$$

**Action on cyclic polynomials.** Now let us continue with cyclic polynomials. Here we will meet some issues in the non-linear local action that are worth paying attention to, as they will come back as actual problems for the non-linear bi-local action.

We start by introducing the cyclic shift operator U which cyclically shifts all fields by one site to the left. In other words, in a monomial of length $n$, it maps site $k$ to site $k-1$ for $k = 2,\dots,n$ and site 1 to site $n$,

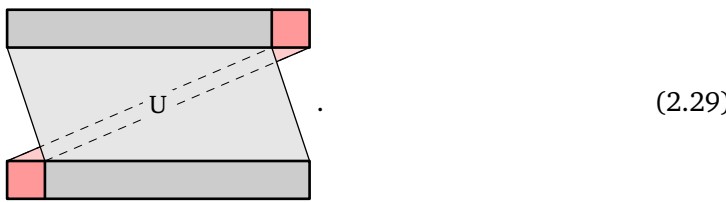

$$. \tag{2.29}$$

The shift operator allows us to define a cyclic polynomial $\mathscr{O}$ as a closed polynomial that is invariant under cyclic shifts

$$U\mathscr{O} = \mathscr{O}. \tag{2.30}$$

Due to cyclic symmetry, it makes sense to define the expansion of a cyclic polynomial $\mathscr{O}$ with non-trivial symmetry factors[7]

$$\mathscr{O} = \sum_n \frac{1}{n} \mathscr{O}_{[n]}. \tag{2.31}$$

Next, we consider the action of a local generator J on a cyclic polynomial $\mathscr{O}$, i.e. a closed polynomial obeying $U\mathscr{O} = \mathscr{O}$. The above symmetry factors produce some curious prefactors for the resulting expansion

$$\text{J}\cdot\mathscr{O} = \sum_n \frac{1}{n}(\text{J}\cdot\mathscr{O})_{[n]} \qquad \text{with} \qquad (\text{J}\cdot\mathscr{O})_{[n]} = \sum_{m=0}^{n} \frac{n}{n-m} \text{J}_{[m]}\cdot\mathscr{O}_{[n-m]}. \tag{2.32}$$

To determine the coefficients $(\text{J}\cdot\mathscr{O})_{[n]}$, we might treat cyclic polynomials as if they were open polynomials because the local action does not pay attention to ordering

$$(\text{J}\cdot\mathscr{O})_{[n]} \overset{?}{=} \sum_{m=0}^{n} \sum_{k=1}^{n-m} \frac{n}{n-m} \text{J}_{[m],k}\mathscr{O}_{[n-m]}. \tag{2.33}$$

However, this assignment is problematic because the resulting expansion coefficients are not cyclic even though $\mathscr{O}$ is. One can easily show this by considering the commutation relations for the shift operator

$$\text{J}_{[m],k}U\mathscr{O}_{[n]} = \begin{cases} U\text{J}_{[m],k+1}\mathscr{O}_{[n]} & \text{for } k \neq n, \\ U^{m+1}\text{J}_{[m],1}\mathscr{O}_{[n]} & \text{for } k = n. \end{cases} \tag{2.34}$$

The special case $k = n$ requires an adjustment due to the change of length of the non-linear operator:

$$\tag{2.35}$$

To resolve cyclicity violation, note that field polynomials in a gauge theory maintain their trace structure when acted upon by a local operator. Therefore we should project the result to the cyclic component, let us do this by introducing an operator 'tr' as the cyclic projection

$$\text{tr}\,\mathscr{O}_{[n]} := \sum_{k=1}^{n} \frac{1}{n} U^k \mathscr{O}_{[n]}. \tag{2.36}$$

Then the expansion coefficients of the local action are given by the manifestly cyclic expression

$$(\text{J}\cdot\mathscr{O})_{[n]} = \text{tr} \sum_{m=0}^{n} \sum_{k=1}^{n-m} \frac{n}{n-m} \text{J}_{[m],k}\mathscr{O}_{[n-m]} = \sum_{m=0}^{n} \sum_{k=1}^{n} U^k \text{J}_{[m],1}\mathscr{O}_{[n-m]}$$

$$= \sum_{m=0}^{n} \left[ \sum_{k=1}^{n-m} \text{J}_{[m],k}\mathscr{O}_{[n-m]} + \sum_{k=1}^{m} U^k \text{J}_{[m],1}\mathscr{O}_{[n-m]} \right]. \tag{2.37}$$

---

[7]The symmetry factors are useful as they naturally cancel many factors due to combinatorics, but one has to pay attention that $\mathscr{O}_{[n]}$ merely represents the $n$-field contribution of a cyclic polynomial $\mathscr{O}$ up to a factor of $n$.

The second expression on the first line is manifestly cyclic, and the prefactor has been cancelled by the factor $1/n$ due to cyclic symmetrisation in (2.36) and due to the equivalence of all $n-m$ summands under cyclic projection. The second line is an alternative presentation where we have a separated the $n-m$ naive actions on the individual fields from terms where the result of the action on a field extends past the closure at sites $n$ and $1$,

$$\sum_{k=1}^{n-m} \boxed{J_k} + \sum_{k=1}^{m} \boxed{J_1} \quad U^k \;.\tag{2.38}$$

The latter is a somewhat unintuitive effect of non-linear representations on cyclic polynomials.

Calculations can become somewhat cumbersome when we handle cyclic projections explicitly. It therefore makes sense to work modulo cyclic identifications using the equivalence relation '$\simeq$'. In that case we readily find

$$(J \cdot \mathcal{O})_{[n]} \simeq \sum_{m=0}^{n} n\, J_{[m],1}\, \mathcal{O}_{[n-m]}.\tag{2.39}$$

The prefactor even cancels upon reassembling all expansion coefficients

$$J \cdot \mathcal{O} \simeq \sum_{n} \sum_{m=0}^{n} J_{[m],1}\, \mathcal{O}_{[n-m]} \simeq \sum_{n,m} J_{[m],1}\, \mathcal{O}_{[n]}.\tag{2.40}$$

When acting with further local operators, one has to pay attention to proper prefactors and cyclic symmetrisations at intermediate stages. For example, one finds

$$J^a \cdot J^b \cdot \mathcal{O} \simeq \sum_{n,m,p} \sum_{j=1}^{n+m} J^a_{[p],1}\, U^j\, J^b_{[m],1}\, \mathcal{O}_{[n]} \simeq \sum_{n,m,p} \sum_{j=1}^{n+m} J^a_{[p],j}\, J^b_{[m],1}\, \mathcal{O}_{[n]}.\tag{2.41}$$

From here it is straight-forward to see that the action on cyclic polynomials obeys the same commutation relations (2.25, 2.26) as for open polynomials, and therefore also forms a proper representation of the symmetry algebra.

## 2.4 Non-linear bi-local actions

Let us now turn our attention to bi-local generators. We shall denote a bi-local combination of two generators $J^a$ and $J^b$ by the composition '$\otimes$' as $J^a \otimes J^b$. Qualitatively, $J^a \otimes J^b$ describes a bi-local action where $J^a$ acts only on sites to the left of the location where $J^b$ acts. According to (2.14), the appropriate notation for the level-one generator $\widehat{J}$ therefore reads

$$\widehat{J} = J^{(1)} \otimes J^{(2)}.\tag{2.42}$$

As explained earlier, the bi-local action is accompanied by some local action which is needed to complete the definition in the limit of coincident insertions. Here we assume that for every bi-local action $J^a \otimes J^b$ there will be a suitable choice for the local part.[8] The local action can be isolated as the action on a single field, $(J^a \otimes J^b) \cdot Z$.

Noting that the product of two local operators $J^a$ and $J^b$ has a bi-local structure, it is natural to demand the relationship

$$J^a \otimes J^b + J^b \otimes J^a = \tfrac{1}{2} J^a J^b + \tfrac{1}{2} J^b J^a.\tag{2.43}$$

---

[8]Note that the choice of local part may not be unique, but by construction the difference between two choices $J^a \otimes J^b$ and $(J^a \otimes J^b)'$ is a local operator which we know how to handle.

In particular, the local parts should be defined in consistency with this relationship. This implies that a symmetric combination of generators $J^a$ and $J^b$ is equivalent to a product of two local generators. Therefore, we may restrict to anti-symmetric combinations, which is enforced by the anti-symmetric composition '$\wedge$'

$$J^a \wedge J^b := J^a \otimes J^b - J^b \otimes J^a. \tag{2.44}$$

However, for the level-one generator $\widehat{J} = J^{(1)} \otimes J^{(2)}$ anti-symmetry is implied by Sweedler's notation, and there is no need for explicit anti-symmetrisation (note that $J^{(1)} \wedge J^{(2)} = 2\widehat{J}$).

**Open polynomials.** As the constituent generators $J^a$ and $J^b$ of $J^a \otimes J^b$ will typically be non-linear representations, the bi-local action $J^a \otimes J^b$ should also be non-linear. The generalisation of the non-linear action to bi-local generators was proposed in [8], and it reads straightforwardly

$$(J^a \otimes J^b) \cdot \mathscr{X} := \sum_{n,m} \sum_{k=1}^{n} (J^a \otimes J^b)_{[m],k} \mathscr{X}_{[n]} + \sum_{n,m,p} \sum_{k=2}^{n} \sum_{j=1}^{k-1} J^a_{[p],j} J^b_{[m],k} \mathscr{X}_{[n]}. \tag{2.45}$$

The first term is a local action while the second term is the bi-local action.

Note that we chose to keep the bi-local terms in our definition (2.45) well-separated. One might just as well include terms where $J^a$ acts on the output of $J^b$ or vice versa. However these contributions would amount to local terms, and they could be compensated by a change of the local term $(J^a \otimes J^b)_{\text{loc}}$. In that sense one may view the local term as a regulator for the short-distance limit of the bi-local terms.

**Cyclic polynomials.** In [9] the action of a bi-local generator on a cyclic polynomial was proposed that is suitable to express invariance of the action

$$
\begin{aligned}
(J^a \wedge J^b) \cdot \mathscr{O} :\simeq & \sum_{n,m} (J^a \wedge J^b)_{[m],1} \mathscr{O}_{[n]} \\
& + \sum_{n,m,l} \sum_{k=2}^{n} \frac{2k-n-2}{n+m+l} J^a_{[m],k+l} J^b_{[l],1} \mathscr{O}_{[n]} \\
& + \sum_{n,m,l} \sum_{k=1}^{l+1} \frac{2k-l-2}{n+m+l} J^a_{[m],k} J^b_{[l],1} \mathscr{O}_{[n]} \\
& - \sum_{n,m,l} \sum_{k=1}^{m+1} \frac{2k-m-2}{n+m+l} J^b_{[l],k} J^a_{[m],1} \mathscr{O}_{[n]}.
\end{aligned}
\tag{2.46}
$$

Here, the first line contains local terms and the second line describes the non-overlapping bi-local contributions. The last two lines consist of overlapping bi-local terms which turned out to be essential for making the action properly invariant. They can be viewed as the appropriate short-distance regulator for cyclic polynomials. Even though the overlapping terms act in a local fashion, we shall lump them together with the bi-local terms in the restriction $(J^a \wedge J^b)_{\text{biloc}}$; the local part $(J^a \wedge J^b)_{\text{loc}}$ will consist of the terms in the first line of (2.46) only.

Curiously, the bi-local terms have a set of unusual prefactors. In particular, they depend explicitly on the length $n$ of the cyclic polynomial they act upon. In fact, the denominator $n+m+l$ (which equals the length of the resulting polynomial) will turn out to be troublesome when commuting operators because the relevant length is measured at different stages of the calculation and the corresponding denominators will not match up. This is a purely non-linear effect of which we will see some examples further below; fortunately, the mismatching terms can be eliminated by suitable restrictions.

A related concern is gauge covariance: while the above action on open polynomials (2.45) is manifestly gauge-covariant (provided that the local contributions are), this is not the case for the expression (2.46) because the non-linear combinations needed for gauge covariance are apparently upset by different prefactors for different terms. We shall return to and resolve this issue in Sec. 3.3.

The above bi-local action (2.46) was derived in [9] under three conditions: manifest anti-symmetry of $J^a$ and $J^b$, invariance of the cyclic polynomial $\mathcal{O}$ under $J^a$ and $J^b$ as well as commutativity of $J^a$ and $J^b$. The superconformal symmetries $J^a$ and the action $\mathcal{S}$ of integrable planar gauge theories such as $\mathcal{N} = 4$ sYM satisfy these requirements. Let us comment on these conditions.

The first condition is reflected by the fact that the definition (2.46) only makes reference to the anti-symmetric bi-local operator $J^a \wedge J^b$. As mentioned earlier, this is not actually a restriction because the symmetric part merely corresponds to the subsequent action of two local generators, (2.43). Nevertheless, the symmetric and anti-symmetric parts do not mix well for cyclic polynomials, and we shall restrict to manifestly anti-symmetric bi-local operators.

The second condition reads

$$J^a \cdot \mathcal{O} = J^b \cdot \mathcal{O} = 0. \tag{2.47}$$

In this work, we will also act on operators $\mathcal{O}$ other than the action $\mathcal{S}$ such that the constituent generators do not annihilate $\mathcal{O}$. In that case, it may seem natural to supplement (2.46) with terms proportional to $J^{(1)} \cdot \mathcal{O}$ or $J^{(2)} \cdot \mathcal{O}$. Further admissible terms are of the form

$$\sum_{n,m,l} a_{n+l,m} \sum_{k=1}^{n+l} J^a_{[m],k} J^b_{[l],1} \mathcal{O}_{[n]} - \sum_{n,m,l} a_{n+m,l} \sum_{k=1}^{n+m} J^b_{[l],k} J^a_{[m],1} \mathcal{O}_{[n]}, \tag{2.48}$$

with some arbitrary coefficients $a_{n,m}$. However, they clearly cannot change the form of the action significantly, and we will stick to the expression (2.46) to enable us to formulate concrete statements. Allowing for $a_{n,m} \neq 0$ will turn out unnecessary in our analysis.

Third, we have also assumed that the constituent operators $J^a$ and $J^b$ commute exactly

$$[J^a, J^b] = 0. \tag{2.49}$$

Literally, this statement will hold only for very specific choices of $J^a$ and $J^b$. In fact, it suffices that a linear combination $J^{(1)} \otimes J^{(2)}$ of bi-local terms using Sweedler's notation satisfies the requirement

$$[J^{(1)}, J^{(2)}] = 0. \tag{2.50}$$

While this requirement might in principle be relaxed as the other two, this would typically have direr consequences. For the purposes of this article, we shall only consider bi-local operators satisfying (2.50).

# 3 Issues of bi-local algebra

In this section we will comment on issues related to the commutation algebra of bi-local generators with local generators, and the role of gauge transformations. We will see that closure of the Yangian algebra requires to introduce novel bi-local gauge transformations which serve as additional symmetries of the planar gauge theory action.

### 3.1   Bi-local commutators

In the following we consider the commutator of a generic bi-local generator $J^a \otimes J^b$ with another local generator $J^c$. We will derive the expression by acting on an open polynomial $\mathcal{X}$. Subsequently, we will discuss the algebra for cyclic polynomials which bears some complications.

**Open polynomials.**   We first act with the commutator of bi-local and local operators on a generic open polynomial $\mathcal{X}$ using the expressions (2.22,2.23) and (2.45). The calculation is straight-forward but it requires some patience due to the various non-linear terms. By considering the bi-local terms we find that all non-linear contributions combine nicely into commutators of local generators (2.25,2.26). We find the unsurprising result

$$\left[J^c, J^a \otimes J^b\right] \cdot \mathcal{X} = \left([J^c, J^a] \otimes J^b\right) \cdot \mathcal{X} + \left(J^a \otimes [J^c, J^b]\right) \cdot \mathcal{X} + \text{local}. \tag{3.1}$$

The remaining local contributions to the relation depend on the precise definition of the local terms in the three bi-local operators. The local terms from the commutator can be expressed as

$$
\begin{aligned}
\left[J^c, J^a \otimes J^b\right]_{[n],j} =& \sum_{m=0}^{n}\sum_{k=1}^{m+1} J^c_{[n-m],k+j-1}(J^a \otimes J^b)_{[m],j} - \sum_{m=0}^{n}\sum_{k=1}^{m+1}(J^a \otimes J^b)_{[n-m],k+j-1}J^c_{[m],j} \\
&- \sum_{m=1}^{n}\sum_{l=0}^{n-m}\sum_{k=2}^{m+1}\sum_{i=1}^{k-1} J^a_{[n-m-l],i+j-1}J^b_{[l],k+j-1}J^c_{[m],j}.
\end{aligned}
\tag{3.2}
$$

The terms on the first line correspond to the ordinary commutator of the local parts, cf. (2.26), whereas the second term originates from both components of the bi-local generator acting on the non-linear result of the local generator.

**Cyclic polynomials.**   The commutator algebra for the bi-local action on a cyclic polynomial $\mathcal{O}$ is hardly as straight-forward as the open polynomial counterpart. According to (3.1) one would expect

$$\left[J^c, J^a \wedge J^b\right] \cdot \mathcal{O} \simeq \left([J^c, J^a] \wedge J^b\right) \cdot \mathcal{O} + \left(J^a \wedge [J^c, J^b]\right) \cdot \mathcal{O} + \text{local}. \tag{3.3}$$

Unfortunately, this relationship is very hard to evaluate for a variety of reasons: The intermediate expressions involve three generators $J^a$, $J^b$ and $J^c$ which can act on different positions within the polynomial. The insertion of generators commutes with each other unless they overlap. Due to cyclic symmetry, only the relative insertion points matter and cyclic symmetry allows to cyclically permute (non-overlapping) generators. In addition, each of the three generators as well as the polynomial consists of terms of different length. Altogether this amounts a six-fold sum with non-trivial boundary conditions for each term. Moreover, the validity of the statement depends on certain algebraic constraints, which are equally hard to spot in the residual terms. In fact, almost all cancellations between terms are due to these constraints.

In order to streamline the calculation, we shall address a bi-local generator $Q \otimes Q$ composed from two equal fermionic generators $Q$. Anti-symmetry of the expression is then manifest. The action (2.46) on a cyclic polynomial $\mathcal{O}$ reduces somewhat to

$$
\begin{aligned}
(Q \otimes Q) \cdot \mathcal{O} \simeq & \sum_{n,m}(Q \otimes Q)_{[m],1}\mathcal{O}_{[n]} \\
&+ \sum_{n,m,l}\sum_{k=2}^{n}\frac{2k-n-2}{2(n+m+l)}Q_{[m],k+l}Q_{[l],1}\mathcal{O}_{[n]} \\
&+ \sum_{n,m,l}\sum_{k=1}^{l+1}\frac{2k-l-2}{n+m+l}Q_{[m],k}Q_{[l],1}\mathcal{O}_{[n]}.
\end{aligned}
\tag{3.4}
$$

Note that the specialisation to Q⊗Q is in fact not a restriction. Due to linearity of all expressions in each generator $J^a$ and $J^b$, we can recover a corresponding relationship for $J^a \wedge J^b$ by means of a replacement

$$\ldots Q \ldots Q \ldots \to \ldots J^a \ldots J^b \ldots - \ldots J^b \ldots J^a \ldots . \tag{3.5}$$

This replacement is to be understood for every line and every term of the calculation when read from left to right. Any signs due to exchange statistics of the fermionic generators Q is reflected by the explicit anti-symmetry of $J^a$ and $J^b$. Moreover, we can now denote the third generator $J^c$ by J and avoid some of the index structure while paying attention to the fermionic nature of the generators Q.

The statement we need to show thus reduces (3.3) to

$$[J, Q \otimes Q] \cdot \mathscr{O} \simeq \big([J,Q] \wedge Q\big) \cdot \mathscr{O}. \tag{3.6}$$

Here we have assumed that the local terms in (3.3) have been absorbed into the definition of the resulting bi-local generator $[J,Q] \wedge Q$ which is defined by the corresponding relation (3.1) on open polynomials $\mathscr{X}$ without local term

$$[J, Q \otimes Q] \cdot \mathscr{X} = \big([J,Q] \wedge Q\big) \cdot \mathscr{X}. \tag{3.7}$$

Hence, the local contributions to $[J,Q] \wedge Q$ are completely defined by (3.2). By means of a lengthy calculation, we find[9][10]

$$
\begin{aligned}
[J, Q \otimes Q] \cdot \mathscr{O} &\simeq J \cdot (Q \otimes Q) \cdot \mathscr{O} - (Q \otimes Q) \cdot J \cdot \mathscr{O} \\
&\simeq \big([J,Q] \wedge Q\big) \cdot \mathscr{O} \\
&\quad + \sum_{n,m,p} \sum_{k=2}^{n} \Big( \frac{2k-n-2}{n+m+p} - \frac{2k-n-2}{n+m} \Big) J_{[p],k+m} \big[ Q_{[m],1}(Q \cdot \mathscr{O})_{[n]} - \tfrac{1}{2}\{Q,Q\}_{[m],1}\mathscr{O}_{[n]} \big] \\
&\quad + \sum_{n,m,p} \sum_{k=1}^{m+1} \Big( \frac{2k-m-2}{n+m+p} - \frac{2k-m-2}{n+m} \Big) J_{[p],k} \big[ Q_{[m],1}(Q \cdot \mathscr{O})_{[n]} - \tfrac{1}{2}\{Q,Q\}_{[m],1}\mathscr{O}_{[n]} \big] \\
&\quad - \sum_{n,m,p} \sum_{k=1}^{p+1} \frac{2k-p-2}{n+m+p} \big[ Q_{[m],k}J_{[p],1}(Q \cdot \mathscr{O})_{[n]} - \tfrac{1}{2}\{Q,Q\}_{[m],k}J_{[p],1}\mathscr{O}_{[n]} \big].
\end{aligned}
\tag{3.8}
$$

We have arranged all residual terms as combinations of generators acting on $Q \cdot \mathscr{O}$ and as combinations of generators involving $\{Q,Q\}$ acting on $\mathscr{O}$. If we impose the restrictions (2.47,2.49) used in deriving the form of the non-linear bi-local action on cyclic polynomials (3.3), namely $Q \cdot \mathscr{O} \simeq 0$ and $\{Q,Q\} = 0$, we find that the commutator algebra comes out as expected for cyclic polynomials. In other words, (2.46) indeed defines a proper algebraic representation of the bi-local operator $J^a \wedge J^b$ on cyclic polynomials $\mathscr{O}$ which is compatible with the corresponding relation (3.1) on open polynomials provided that the constraints (2.47,2.49) hold.

## 3.2 Yangian algebra

As already discussed in Sec. 2.3, the level-zero algebra closes only modulo field-dependent gauge transformations. Consider for example the momentum generator $P_\mu$ for which we can express the resulting gauge transformations in generality as

$$[P_\mu, J] = [P_\mu, J]_{\mathfrak{g}} + G[J \cdot A_\mu]. \tag{3.9}$$

---

[9]Note that all remaining terms vanish if J has a purely linear action, i.e. for $p = 0$, hence they are clearly effects of non-linear actions.

[10]The remaining terms are somewhat reminiscent of the terms in $(J \wedge Q) \cdot Q \cdot \mathscr{O}$.

Here, the first term represents the result of the straight level-zero algebra $\mathfrak{g}$ with the ideal of gauge transformations quotiented out. The second term in the algebra relations specifying a gauge transformation is largely unproblematic on its own because it clearly disappears when acting on gauge-invariant objects. For instance, if both $P_\mu$ and J are symmetries of the action, so is their commutator. The additional gauge transformation term then does not make a difference as the action is gauge-invariant by construction.

The appearance of gauge transformations in the level-zero algebra naturally has an impact on the algebra involving level-one generators (2.12) and (2.13). At the very least, one would expect gauge transformations to appear there as well, however, the situation turns out to be more involved. Let us consider the bi-local part of the level-one momentum generator

$$\widehat{P} = P^{(1)} \otimes P^{(2)}. \tag{3.10}$$

A commutator with the level-zero momentum generator $P_\mu$ yields (3.1)

$$[P_\mu, \widehat{P}] = G[P^{(1)} \cdot A_\mu] \wedge P^{(2)} + \text{local}. \tag{3.11}$$

All the regular non-gauge terms cancel as they should within the Yangian algebra, see (2.12), but the contributions from gauge transformations remain. However, the resulting bi-local action is not a gauge transformation as such; it merely involves gauge transformations alongside level-zero transformations. Consequently, the action of the commutator cannot be expected to vanish simply on gauge-invariant objects, and we need to understand in what sense it can be considered trivial. Conveniently, some substantial simplifications come about due to the partial gauge transformation, and we will show in the following Sec. 3.3 in more generality that the resulting bi-local term annihilates gauge-invariant objects which are invariant under level-zero transformations at the same time. As the latter properties apply to the action $\mathscr{S}$ by construction (irrespectively of whether the complete Yangian algebra is a symmetry or not) the resulting bi-local term must be part of an algebraic ideal, which can be discarded to recover the plain Yangian algebra.

This property puts us in a good position to argue that the adjoint property (2.12) holds modulo (bi-local) gauge transformations, and that the non-gauge level-one generators transform in the adjoint representation of the non-gauge level-zero generators. The same should apply to the Serre relations (2.13): modulo (multi-local) gauge transformations, one can expect to find precisely the Yangian relations (2.13) because the non-gauge terms follow (2.13). Nevertheless it would be desirable to confirm explicitly all the Yangian commutation relations (2.12) and (2.13) and to derive the concrete decorations due to gauge transformations.

### 3.3 Bi-local gauge symmetries

We have seen above that commutators at level one yield terms of the kind $G[\mathscr{X}] \otimes J$, where $G[\mathscr{X}]$ is a gauge transformation with gauge parameter $\mathscr{X}$ and J is some level-zero transformation.

**Definition.** We will now discuss how such a bi-local generator based on a sequence of fields $\mathscr{X}$ acts on a generic sequence $Z_1 \cdots Z_n$ of $n$ fields without derivatives. Using the special form of a gauge transformation, we can immediately write the (purely bi-local) action as

$$
\begin{aligned}
(G[\mathscr{X}] \otimes J)_{\text{biloc}} \cdot (Z_1 \cdots Z_n) &= \sum_{k=1}^{n} \mathscr{X} Z_1 \cdots Z_{k-1} (J \cdot Z_k) Z_{k+1} \cdots Z_n \\
&\quad - \sum_{k=1}^{n} Z_1 \cdots Z_{k-1} \mathscr{X} (J \cdot Z_k) Z_{k+1} \cdots Z_n.
\end{aligned}
\tag{3.12}
$$

Here the first term is a bi-local bulk-boundary term while the second one is purely local. Interestingly, we can remove the latter completely by defining the local action of $G[\mathscr{X}] \otimes J$ as

$$(G[\mathscr{X}] \otimes J) \cdot Z := \mathscr{X} \, J \cdot Z. \tag{3.13}$$

The combined bi-local and local action can then be written as the action of J dressed by the sequence $\mathscr{X}$

$$(G[\mathscr{X}] \otimes J) \cdot (Z_1 \cdots Z_n) = \mathscr{X} \, J \cdot (Z_1 \cdots Z_n). \tag{3.14}$$

So far, we have ignored derivative terms. It turns out that the local and bi-local terms neatly conspire to yield a consistent expression on (non-linear) covariant derivatives

$$(G[\mathscr{X}] \otimes J) \cdot (\nabla_\mu Z) = \mathscr{X} \, J \cdot (\nabla_\mu Z). \tag{3.15}$$

Therefore the form of the action on polynomials (3.14) also holds in the presence of covariant derivatives.

Evidently, the bi-local generator $J \otimes G[\mathscr{X}]$ with the opposite ordering of constituent generators behaves analogously:

$$(J \otimes G[\mathscr{X}]) \cdot Z := -J \cdot Z \, \mathscr{X} \quad \Longrightarrow \quad (J \otimes G[\mathscr{X}]) \cdot (Z_1 \cdots Z_n) = -J \cdot (Z_1 \cdots Z_n) \mathscr{X}. \tag{3.16}$$

For the anti-symmetric combinations $G[\mathscr{X}] \wedge J$ appearing within level-one commutators, one thus obtains

$$(G[\mathscr{X}] \wedge J) \cdot (Z_1 \cdots Z_n) = \big\{ \mathscr{X}, J \cdot (Z_1 \cdots Z_n) \big\}. \tag{3.17}$$

With this form, the remaining local term in (3.11) can now be fixed by a direct computation as

$$[P_\mu, \widehat{P}] = G[P^{(1)} \cdot A_\mu] \wedge P^{(2)} + G[\widehat{P} \cdot A_\mu]. \tag{3.18}$$

**Symmetry.** A relevant observation is that the above bi-local generators preserve the form of the polynomial on which they act. If we apply one of them to the equations of motion $\check{Z} \approx 0$ corresponding to some field $Z$, we find

$$(G[\mathscr{X}] \wedge J) \cdot \check{Z} = \big\{ \mathscr{X}, J \cdot \check{Z} \big\}. \tag{3.19}$$

Supposing that J is a symmetry of the equations of motion, we know that $J \cdot \check{Z} \approx 0$ and consequently $(G[\mathscr{X}] \wedge J) \cdot \check{Z} \approx 0$. This is a necessary requirement for $G[\mathscr{X}] \wedge J$ being a symmetry, cf. the discussion in [9].

Let us therefore check whether any gauge-invariant action $\mathscr{S}$ invariant under a local generator J is also invariant under the bi-local transformation $G[\mathscr{X}] \wedge J$ by means of the bi-local action (2.46) on $\mathscr{S}$. Expanding the gauge transformations in terms of an operator $I[\mathscr{X}]$ to insert a sequence of fields $\mathscr{X}$ between any two fields of the polynomials and collapsing some telescoping sums, we find

$$(G[\mathscr{X}] \wedge J) \cdot \mathscr{S} \simeq 2 I[\mathscr{X}] \cdot (J \cdot \mathscr{S}) + 2 I[J \cdot \mathscr{X}] \cdot \mathscr{S}. \tag{3.20}$$

Here the first term is analogous to (3.17), and it vanishes if J is a symmetry of the action. The second term inserts the expression $J \cdot \mathscr{X}$ at all places in the action, and one can hardly expect it to be a symmetry. To make it vanish, $J \cdot \mathscr{X}$ must be zero, and according to (2.19) this is the case if the two constituent operators $G[\mathscr{X}]$ and J commute as they should according to the constraint (2.49). Interestingly, this term vanishes for the residual bi-local transformations $G[P^{(1)} \cdot A_\mu] \wedge P^{(2)}$ arising from the level-one Yangian algebra (3.11) because $P^{(2)} \cdot P^{(1)} \cdot A_\mu = 0$. Therefore the bi-local gauge transformation is a symmetry of our model.

**Gauge covariance.** Another relevant question which we have not yet addressed is gauge covariance of the bi-local action (2.46) of $J^a \otimes J^b$ on a cyclic polynomial $\mathcal{O}$. By specialising (3.3) to $J^c = G[\mathcal{X}]$ we find

$$\big[G[\mathcal{X}], J^a \wedge J^b\big]\cdot\mathcal{O} \simeq \big([G[\mathcal{X}], J^a] \wedge J^b\big)\cdot\mathcal{O} + \big(J^a \wedge [G[\mathcal{X}], J^b]\big)\cdot\mathcal{O}, \qquad (3.21)$$

provided that the constraints (2.47,2.49) hold as they should, namely $J^a\cdot\mathcal{O} = J^b\cdot\mathcal{O} = [J^a, J^b] = 0$. We transform further using the commutators of gauge transformations (2.19,3.20),

$$\big[G[\mathcal{X}], J^a \wedge J^b\big]\cdot\mathcal{O} \simeq -2I[J^a\cdot\mathcal{X}]\cdot(J^b\cdot\mathcal{O}) + 2I[J^b\cdot\mathcal{X}]\cdot(J^a\cdot\mathcal{O}) + 2I\big[[J^a, J^b]\cdot\mathcal{X}\big]\cdot\mathcal{O}. \qquad (3.22)$$

All resulting terms vanish due to the prerequisites for $J^a$, $J^b$ and $\mathcal{O}$. This shows that the bi-local action (2.46) is gauge-covariant in the sense that its result on a gauge-invariant cyclic polynomial is again gauge-invariant subject to the constraints (2.47,2.49).

# 4 Gauge fixing

In order to consistently quantise a gauge theory it is necessary to fix a gauge such that the kinetic terms in the action become invertible. In many cases, gauge fixing breaks some symmetries of the theory, e.g. conformal symmetry, because the gauge-fixing terms or constraints transform non-trivially under the symmetry. The challenge is thus to show that the violations of symmetry do not affect physical quantities.

## 4.1 Ghosts and BRST symmetry

We will use the Faddeev–Popov procedure to fix the gauge which introduces some additional propagating ghost fields. In this framework, BRST symmetry is typically used to select physical processes and to show that the unphysical degrees of freedom do not contribute to them.

Towards establishing the Yangian as a symmetry of the quantum theory and of quantum observables, it is important to show that the gauge-fixing procedure does not spoil the invariance of the action. We thus need to specify how the Yangian is represented on the ghost fields and whether the representation on physical terms receives additional contributions from the ghost fields. We should then show that the gauge-fixed action remains Yangian-invariant in a suitable sense.

We choose to work in the standard Lorentz-invariant family of gauges given by the gauge-fixing terms[11]

$$\mathcal{S}_{\text{gf}} = \int \mathrm{d}x^d \, \mathrm{tr}\big[\nabla_\mu C \, \partial^\mu \bar{C} - B \, \partial^\mu A_\mu + \tfrac{1}{2}\xi B^2\big]. \qquad (4.1)$$

The fields $\bar{C}$ and $C$ are the Faddeev–Popov ghosts (anti-commuting scalars), and $B$ is an auxiliary bosonic scalar field. As all the other fields of the theory, the new fields are $N_c \times N_c$ matrices. The action $\mathcal{S}$ of the gauge-fixed model is the sum of the original action $\mathcal{S}_0$ and the gauge-fixing terms in $\mathcal{S}_{\text{gf}}$

$$\mathcal{S} = \mathcal{S}_0[Z] + \mathcal{S}_{\text{gf}}[Z, C, \bar{C}, B]. \qquad (4.2)$$

---

[11]This gauge-fixing term should be applicable to arbitrary gauge theories in any number of spacetime dimensions $d$. Note that the constant $\xi$ carries mass dimension $d-4$, so it is dimensionless only for $d=4$ and the term should be renormalisable for $d < 4$. One can drop the $B^2$ term altogether such that the auxiliary field $B$ will become a Lagrange multiplier enforcing the Lorenz gauge $\partial^\mu A_\mu = 0$ tightly. In any case, the term plays a minor role for our investigations as it will drop out from almost all calculations at the very beginning.

Due to the appearance of the bare gauge field $A$ and non-covariant partial derivatives $\partial$, the terms $\mathscr{S}_{\text{gf}}$ break the gauge invariance of the theory. A remnant thereof however survives; it is known as the BRST symmetry and it is generated by the action of a fermionic generator Q [12] on the fields of the theory

$$\text{Q·}Z = \text{G}[C]\text{·}Z, \qquad \text{Q·}C = CC, \qquad \text{Q·}\bar{C} = iB, \qquad \text{Q·}B = 0, \qquad (4.3)$$

where $Z$ denotes any of the fields of the original model. For $\xi \neq 0$, one might integrate out the auxiliary field $B$ whose equation of motion is algebraic, $B \approx \xi^{-1}\partial^{\mu}A_{\mu}$, but this would obscure some statements about irrelevant contributions.

BRST symmetry has the special property that it squares to zero

$$\text{QQ} = \tfrac{1}{2}\{\text{Q}, \text{Q}\} = 0, \qquad (4.4)$$

and therefore it defines a cohomology. The latter is useful in specifying physical objects which have to be closed and carry ghost number zero. In particular the action is BRST-closed. For the original action $\mathscr{S}_0$ this follows from gauge symmetry,

$$\text{Q·}\mathscr{S}_0 = \text{G}[C]\text{·}\mathscr{S}_0 = 0, \qquad (4.5)$$

and for the gauge-fixing terms $\mathscr{S}_{\text{gf}}$ it follows from BRST-exactness

$$\mathscr{S}_{\text{gf}} = -\text{Q·}\mathscr{K}_{\text{gf}}, \qquad \mathscr{K}_{\text{gf}} = \int \text{d}x^d \, \text{tr}\big[iA_\mu\,\partial^\mu\bar{C} + \tfrac{i}{2}\xi B\bar{C}\big]. \qquad (4.6)$$

In fact, the latter feature is important because BRST-exactness indicates that the gauge-fixing terms effectively do not contribute to physical processes.

## 4.2 Local symmetries

Our investigations are based on the assumption that the original action is invariant under some symmetries J

$$\text{J·}\mathscr{S}_0 = 0. \qquad (4.7)$$

It is clear that gauge fixing breaks some of these symmetries, $\text{J·}\mathscr{S}_{\text{gf}} \neq 0$, so that altogether $\text{J·}\mathscr{S} \neq 0$. In particular, we have argued that the level-zero algebra closes onto gauge transformations which are no longer exact symmetries of the full system. In order to let gauge fixing preserve a symmetry J we need to show at least that the variation of the action is BRST-exact,[13][14]

$$\text{J·}\mathscr{S} = -\text{Q·}\mathscr{K}[\text{J}]. \qquad (4.8)$$

In order to determine the precise form of $\mathscr{K}[\text{J}]$ we need to fix the action of the level-zero generators J on the additional fields $C$, $\bar{C}$ and $B$. A seeming complication is that these fields typically do not form proper multiplets under the level-zero algebra.[15] This complication could

---

[12]We will make signs due to the fermionic statistics of Q explicit. However, we will keep assuming that the generators J are bosonic, so that any signs for fermionic generators J and $\widehat{\text{J}}$ (e.g. in commutators with Q) are implicit and need to be inserted manually.

[13]In this definition, the natural sign assignment due to statistics reads $\text{J·}\mathscr{S}_{\text{gf}} = -(-1)^{|\text{J}|}\text{Q·}\mathscr{K}[\text{J}]$. It follows by substitution of $\mathscr{S}_{\text{gf}} = -\text{Q·}\mathscr{K}_{\text{gf}}$. and by the assumption that (qualitatively) $\mathscr{K}[\text{J}] \sim \text{J}\mathscr{K}_{\text{gf}}$. Then this matches with the natural sign due to permutation of Q and J: $-\text{JQ·}\mathscr{K}_{\text{gf}} \sim -(-1)^{|\text{J}|}\text{QJ·}\mathscr{K}_{\text{gf}}$.

[14]Note that the additional term on the r.h.s. spoils the on-shell invariance of the equations of motion $\check{Z} \approx 0$ such that $\text{J·}\check{Z} \not\approx 0$. However, there will be some well-prescribed terms involving $\mathscr{K}[\text{J}]$ and Q to compensate the remaining terms.

[15]In particular, this is rather evident in a supersymmetric theory when the level-zero algebra includes supersymmetry.

perhaps be resolved in particular situations by adding further unphysical fields to complete the multiplets, but it would inevitably be a rather complicated solution. The convenient alternative is to declare all these fields singlets under all level-zero generators. In other words, we declare the level-zero representation on the unphysical fields to be trivial,[16] [17]

$$\mathrm{J}{\cdot}C = \mathrm{J}{\cdot}\bar{C} = \mathrm{J}{\cdot}B = 0. \tag{4.9}$$

Furthermore, we do not modify the level-zero representation of the original fields of the model so that the original part of the action $\mathscr{S}_0$ remains invariant according to (4.7). It also nicely ensures that BRST symmetry commutes with the level-zero symmetry

$$[\mathrm{Q},\mathrm{J}] = 0. \tag{4.10}$$

In this case, it is straight-forward to show using (4.6) that (4.8) holds

$$\mathrm{J}{\cdot}\mathscr{S}_{\mathrm{gf}} = -\mathrm{Q}{\cdot}\mathscr{K}[\mathrm{J}], \tag{4.11}$$

with the compensator $\mathscr{K}[\mathrm{J}]$ given by acting with the level-zero generator J on $\mathscr{K}_{\mathrm{gf}}$

$$\mathscr{K}[\mathrm{J}] = \mathrm{J}{\cdot}\mathscr{K}_{\mathrm{gf}} = i\int \mathrm{d}x^d\,\mathrm{tr}\big[\mathrm{J}{\cdot}A_\mu\,\partial^\mu\bar{C}\big]. \tag{4.12}$$

By construction, gauge fixing breaks gauge symmetry. However, gauge transformations arise from the level-zero algebra, and the full amount of gauge symmetry must be preserved in the same sense as the level-zero symmetries. By extending the action of gauge transformations to the ghost fields as[18]

$$\mathrm{G}[\mathscr{X}]{\cdot}C = \mathrm{G}[\mathscr{X}]{\cdot}\bar{C} = \mathrm{G}[\mathscr{X}]{\cdot}B = 0, \tag{4.13}$$

the gauge generator $\mathrm{G}[\mathscr{X}]$ commutes with the BRST generator Q. The corresponding compensator for gauge invariance of the action $\mathrm{G}[\mathscr{X}]{\cdot}\mathscr{S}_{\mathrm{gf}} = -\mathrm{Q}{\cdot}\mathscr{K}[\mathrm{G}[\mathscr{X}]]$ reads

$$\mathscr{K}\big[\mathrm{G}[\mathscr{X}]\big] = \mathrm{G}[\mathscr{X}]{\cdot}\mathscr{K}_{\mathrm{gf}} = i\int \mathrm{d}x^d\,\mathrm{tr}\big[\mathrm{G}[\mathscr{X}]{\cdot}A_\mu\,\partial^\mu\bar{C}\big] = -\int \mathrm{d}x^d\,\mathrm{tr}\big[\nabla_\mu\mathscr{X}\,\partial^\mu\bar{C}\big]. \tag{4.14}$$

## 4.3 Bi-local symmetries

Before discussing the level-one Yangian symmetries, let us introduce a new class of bi-local generators involving BRST generators Q. This case will be instructive as these generators are considerably simpler than the full level-one Yangian generators. Furthermore, they introduce some additional terms in the gauge-fixed invariance statement. They will also be needed for the level-one Yangian generators and later they will be relevant in formulating identities for quantum correlators due to the level-one symmetries.

---

[16]This includes the curious statement that the unphysical fields carry no momentum, no energy and no angular momentum. However, the assignment is formal and counts only towards the notion and representation of level-zero symmetry. Of course, the fields still depend non-trivially on $x$.

[17]There may be other permissible representations as it should not matter much how the unphysical fields transform in the end.

[18]We assume the gauge transformation parameter $\mathscr{X}$ to be an internal field of the theory (or an open polynomial). For an external field $\Lambda$ the action on the ghost would have to be replaced by $\mathrm{G}[\Lambda]{\cdot}C = [\Lambda,C]$.

**Bi-local BRST symmetry.** We start by recalling that we have introduced bi-local transformations based on gauge transformations in Sec. 3.3. Noting that BRST transformations can be viewed as gauge transformations $Q \sim G[C]$ using the ghost field $C$ as gauge transformation parameter, it is conceivable that $Q \wedge J$ and $Q \otimes Q$ will be further symmetries of the planar gauge theory.[19] We shall call them bi-local BRST symmetries.

We first derive their action on single fields by the condition that the bi-local generators commute with plain BRST symmetry $Q$. This is compatible with the absence of bi-local terms due to the bi-local algebra relations (3.1)

$$[Q, Q \otimes Q] = \{Q, Q\} \wedge Q = 0,$$
$$\{Q, Q \wedge J\} = \{Q, Q\} \wedge J - Q \wedge [Q, J] = 0,$$
(4.15)

which follow from nilpotency of $Q$ and from commutation with $J$. We find the following local contributions,[20]

$$
\begin{array}{ll}
(Q \otimes Q) \cdot Z = \frac{1}{2}[C, Q \cdot Z], & (Q \wedge J) \cdot Z = \{C, J \cdot Z\}, \\
(Q \otimes Q) \cdot C = CCC, & (Q \wedge J) \cdot C = 0, \\
(Q \otimes Q) \cdot B = 0, & (Q \wedge J) \cdot B = 0, \\
(Q \otimes Q) \cdot \bar{C} = 0, & (Q \wedge J) \cdot \bar{C} = 0.
\end{array}
$$

It turns out that $Q \otimes Q$ annihilates any ghost-free gauge-invariant cyclic polynomial via the representation (2.46). For invariance under $Q \wedge J$, the cyclic polynomial must also be invariant under $J$. These statements follow in analogy to (3.20).[21] In particular, the original action is invariant under the bi-local BRST symmetries

$$(Q \otimes Q) \cdot \mathscr{S}_0 \simeq 0, \qquad (Q \wedge J) \cdot \mathscr{S}_0 \simeq 0. \tag{4.16}$$

**Gauge-fixed invariance statements.** To compensate for the broken invariance of the gauge-fixing term $\mathscr{S}_{\mathrm{gf}}$, the procedure from level zero turns out to be insufficient for bi-local generators for the following reason: We need to compensate $(J^a \wedge J^b) \cdot \mathscr{S}_{\mathrm{gf}} = -(J^a \wedge J^b) \cdot Q \cdot \mathscr{K}_{\mathrm{gf}}$. Supposing that $Q$ commutes with $J^a \wedge J^b$ we could rewrite this term as $-Q \cdot (J^a \wedge J^b) \cdot \mathscr{K}_{\mathrm{gf}} = -Q \cdot \mathscr{K}[J^a \wedge J^b]$ and declare the compensator to read $\mathscr{K}[J^a \wedge J^b] = (J^a \wedge J^b) \cdot \mathscr{K}_{\mathrm{gf}}$. Unfortunately, the commutator algebra (3.3) on cyclic polynomials cannot be trusted because $\mathscr{K}_{\mathrm{gf}}$ is not invariant under $J^a$ and $J^b$. For a similar reason, we cannot even be sure that $(J^a \wedge J^b) \cdot \mathscr{S}_{\mathrm{gf}}$ is BRST-exact.

We address the above issue by computing the action of $Q \otimes Q$ on $\mathscr{S}_{\mathrm{gf}} = -Q \cdot \mathscr{K}_{\mathrm{gf}}$. To understand the result better, we shall make no assumptions on the fermionic generator $Q$ and on the precise form of cyclic polynomial $\mathscr{K}_{\mathrm{gf}}$ at first. In a calculation analogous to (3.8) we find

$$(Q \otimes Q) \cdot Q \cdot \mathscr{K}_{\mathrm{gf}} \simeq Q \cdot (Q \otimes Q)_{\mathrm{loc}} \cdot \mathscr{K}_{\mathrm{gf}} + \frac{1}{2}\big(Q \wedge \{Q, Q\}\big) \cdot \mathscr{K}_{\mathrm{gf}} + \frac{1}{2}\big(Q \wedge \{Q, Q\}\big)_{\mathrm{loc}} \cdot \mathscr{K}_{\mathrm{gf}}$$
$$- \frac{1}{3} \sum_{n,m,l,p} \sum_{k=3}^{n} \sum_{j=2}^{k-1} Q_{[m],k+p+l} Q_{[l],j+p} Q_{[p],1} \mathscr{K}_{\mathrm{gf},[n]}. \tag{4.17}$$

Among the resulting terms we find the commutator of $Q$ with $Q \otimes Q$, see (4.15), albeit with a prefactor $^1/_2$. Two other terms represent the purely local contributions from bi-local generators, and there is a tri-local term without overlapping contributions in the second line. Now

---

[19]Note that the BRST generator $Q$ commutes with the level-zero generator $J$ as well as with itself, $\{Q, Q\} = 0$, and thus makes the bi-local generators compatible with cyclicity. Furthermore, the BRST generator $Q$ is fermionic, and thus the bi-local combination $Q \otimes Q$ is anti-symmetric on its own.

[20]There is some freedom in assigning the action on the ghosts $B$ and $\bar{C}$. Such a freedom is related to adding a local generator and thus not interesting. We fix these degrees of freedom to a convenient choice.

[21]The expression (3.20) applied to $Q \otimes Q$ may seem to suggest a non-zero result proportional to $I[Q \cdot C] \cdot \mathscr{S}_0$. However, here one has to pay attention to the action $Q \cdot C = CC$ in the overlapping terms, which is almost a gauge transformation, but only up to a factor of $^1/_2$. An explicit calculation then shows that all terms $I[CC] \cdot \mathscr{S}_0$ vanish.

we have constructed $Q \otimes Q$ such that it commutes with $Q$; this eliminates the two terms involving $\{Q, Q\} \wedge Q$. Furthermore, the tri-local term requires polynomials of length 3 or more, but the compensator $\mathscr{K}_{\mathrm{gf}}$ in (4.6) is purely quadratic. Therefore in our case, the above relation reduces to

$$(Q \otimes Q) \cdot Q \cdot \mathscr{K}_{\mathrm{gf}} \simeq Q \cdot (Q \otimes Q)_{\mathrm{loc}} \cdot \mathscr{K}_{\mathrm{gf}}. \tag{4.18}$$

This shows that the usual form (4.8) of gauge-fixed symmetry relationship holds,

$$(Q \otimes Q) \cdot \mathscr{S}_{\mathrm{gf}} \simeq -Q \cdot \mathscr{K}[Q \otimes Q], \tag{4.19}$$

where the compensator is given by merely the local part of the bi-local generator $Q \otimes Q$ acting on $\mathscr{K}$

$$\mathscr{K}[Q \otimes Q] \simeq (Q \otimes Q)_{\mathrm{loc}} \cdot \mathscr{K} \simeq i \int \mathrm{d}x^d \ \mathrm{tr}\big[(Q \otimes Q) \cdot A_\mu \, \partial^\mu \bar{C}\big]. \tag{4.20}$$

The symmetry statement for the mixed bi-local BRST generator $Q \wedge J$ turns out to be somewhat more involved. We first need to generalise the identity (4.17) to $(Q \wedge J) \cdot Q \cdot \mathscr{K}_{\mathrm{gf}}$. We do this by the same trick employed in Sec. 3.1, and replace all triplets of $Q$ in every term of (4.17) (in their order of appearance) by the following linear combinations,

$$(Q, Q, Q) \rightarrow (J, Q, Q) - (Q, J, Q) + (Q, Q, J). \tag{4.21}$$

Again, we eliminate terms due to $\{Q, Q\} = [Q, J] = 0$ as well as tri-local contributions on $\mathscr{K}_{\mathrm{gf}}$ and find

$$-(Q \wedge J) \cdot Q \cdot \mathscr{K}_{\mathrm{gf}} + (Q \otimes Q) \cdot J \cdot \mathscr{K}_{\mathrm{gf}} \simeq J \cdot (Q \otimes Q)_{\mathrm{loc}} \cdot \mathscr{K}_{\mathrm{gf}} + Q \cdot (Q \wedge J)_{\mathrm{loc}} \cdot \mathscr{K}_{\mathrm{gf}}. \tag{4.22}$$

By rearranging the terms, we can express the relationship as $Q \wedge J$ acting on $\mathscr{S}_{\mathrm{gf}}$,

$$(Q \wedge J) \cdot \mathscr{S}_{\mathrm{gf}} \simeq Q \cdot \mathscr{K}[Q \wedge J] - (Q \otimes Q) \cdot \mathscr{K}[J] + J \cdot \mathscr{K}[Q \otimes Q], \tag{4.23}$$

with the additional compensator defined as before as the action of the local part of $Q \wedge J$

$$\mathscr{K}[Q \wedge J] \simeq (Q \wedge J)_{\mathrm{loc}} \cdot \mathscr{K} \simeq i \int \mathrm{d}x^d \ \mathrm{tr}\big[(Q \wedge J) \cdot A_\mu \, \partial^\mu \bar{C}\big]. \tag{4.24}$$

The physical meaning of the two additional terms in (4.23) as compared to (4.8) remains somewhat obscure at this point. One may argue that it is sufficient that these terms, just like the regular term, are based on the 'lesser' generators $J$ and $Q \otimes Q$ in the hierarchy of all symmetries. Moreover these terms are fully determined in terms of other relations, so they do not introduce any additional degrees of freedom. Most importantly, there exists a compensator $\mathscr{K}[Q \wedge J]$ to formulate an exact symmetry relationship. Later in the investigation of identities for quantum correlation functions we shall find a different justification for the above particular combination of terms.

In conclusion, the bi-local BRST generators are curious additional symmetries of gauge-fixed planar gauge theories with invariance of the gauge-fixed action $\mathscr{S}$ expressed as (4.19, 4.23)

$$\begin{aligned} 0 &\simeq (Q \otimes Q) \cdot \mathscr{S} + Q \cdot \mathscr{K}[Q \otimes Q], \\ 0 &\simeq (Q \wedge J) \cdot \mathscr{S} - Q \cdot \mathscr{K}[Q \wedge J] + (Q \otimes Q) \cdot \mathscr{K}[J] - J \cdot \mathscr{K}[Q \otimes Q]. \end{aligned} \tag{4.25}$$

We have explicitly checked validity of these relations in gauge-fixed planar $\mathcal{N} = 4$ sYM. Moreover, they can be expected to be present in other non-integrable planar gauge symmetries after BRST gauge fixing.

**Yangian symmetry.** Let us now turn our attention to the level-one Yangian generators $\widehat{J}$. We will not change the definition of the local terms in $\widehat{J}$, therefore the assumption of Yangian symmetry of the original action remains valid,

$$\widehat{J} \cdot \mathscr{S}_0 \simeq 0. \tag{4.26}$$

As before, we define the single-field action on ghosts to be trivial,

$$\widehat{J} \cdot C = \widehat{J} \cdot \bar{C} = \widehat{J} \cdot B = 0. \tag{4.27}$$

This assignment ensures that the generator commutes with BRST transformations,

$$[Q, \widehat{J}] = 0. \tag{4.28}$$

Towards obtaining a symmetry statement for the gauge-fixed action, we can again generalise the relationship (4.17) using the trick of Sec. 3.1 for the replacement

$$(Q, Q, Q) \rightarrow (J^{(1)}, J^{(2)}, Q) - (J^{(1)}, Q, J^{(2)}) + (Q, J^{(1)}, J^{(2)}). \tag{4.29}$$

The resulting gauge-fixed invariance relationship turns out to have the form

$$\widehat{J} \cdot \mathscr{S}_{\text{gf}} \simeq -Q \cdot \mathscr{K}[\widehat{J}] - (Q \wedge J^{(2)}) \cdot \mathscr{K}[J^{(1)}] + J^{(1)} \cdot \mathscr{K}[Q \wedge J^{(2)}], \tag{4.30}$$

with the compensator taking the established form

$$\mathscr{K}[\widehat{J}] \simeq \widehat{J}_{\text{loc}} \cdot \mathscr{K}_{\text{gf}} \simeq i \int \mathrm{d}x^d \, \text{tr}\big[\widehat{J} \cdot A_\mu \, \partial^\mu \bar{C}\big]. \tag{4.31}$$

The invariance property of the gauge-fixed action $\mathscr{S}$ therefore reads[22]

$$0 \simeq \widehat{J} \cdot \mathscr{S} + Q \cdot \mathscr{K}[\widehat{J}] + (Q \wedge J^{(2)}) \cdot \mathscr{K}[J^{(1)}] - J^{(1)} \cdot \mathscr{K}[Q \wedge J^{(2)}]. \tag{4.32}$$

This proves that the gauge-fixing procedure does not spoil the Yangian invariance of planar gauge theories. The same should hold for the bi-local generators involving gauge transformations discussed in Sec. 3.3, whose gauge fixing proceeds completely analogously.

## 5  Slavnov–Taylor identities

After having fixed the gauge and formulated statements for Yangian symmetries in the gauge-fixed theory, our next goal is to formulate corresponding identities for quantum correlators, so called Slavnov–Taylor identities. These will eventually reduce to unambiguous identities on the physical contributions, but they may also constrain the unphysical degrees of freedom to some extent.

In the following we will derive and propose the Slavnov–Taylor identities for the various symmetries we have encountered so far. As the level-one generators heavily rely on level-zero and (extended) gauge symmetries, we should start with the simpler kinds of symmetries. Unfortunately, we have no firmly established tools to perform all the required transformations for bi-local generators.[23] Therefore we shall test the proposed identities for correlation functions in the following Sec. 6.

---

[22]The necessity of additional terms in the invariance statement for gauge-fixed ABJM theory was pointed out by Matteo Rosso.

[23]These tools would have to make reference to the planar limit which is essential for the bi-local symmetries to work. We are not aware of a suitable set of tools which specifically address the planar path integral.

## 5.1 Local symmetries

We start with the identities for local symmetries which can be derived as usual from the path integral. We act with a local generator J onto the integrand of the path integral and obtain the identity

$$\left\langle \text{J}\cdot\mathscr{O} + i\text{J}\cdot\mathscr{S}\,\mathscr{O} \right\rangle = 0. \tag{5.1}$$

This identity is due to the fact that J acts as a total variation within the path integral, and it can either hit the operator $\mathscr{O}$ or the implicit phase factor $\exp(i\mathscr{S})$. If we specialise this identity to a BRST generator $\text{J} = \text{Q}$ we find that

$$\left\langle \text{Q}\cdot\mathscr{O} \right\rangle = 0, \tag{5.2}$$

because the action itself is BRST-closed, $\text{Q}\cdot\mathscr{S} = 0$. For the other local generators, however, further terms are needed because the gauge-fixed action is not exactly invariant, but only up to a BRST-exact term (4.8). We can cancel the latter term by adding the identity (5.2) applied to the composite operator $i\mathscr{K}[\text{J}]\mathscr{O}$.

$$\left\langle \text{J}\cdot\mathscr{O} - i\mathscr{K}[\text{J}]\text{Q}\cdot\mathscr{O} + i(\text{J}\cdot\mathscr{S} + \text{Q}\cdot\mathscr{K}[\text{J}])\,\mathscr{O} \right\rangle = 0. \tag{5.3}$$

Here we have combined the two terms constituting the gauge-fixed invariance condition (4.8) for J. By dropping them we arrive at the Slavnov–Taylor identity corresponding to the generator J

$$\left\langle \text{J}\cdot\mathscr{O} - i\mathscr{K}[\text{J}]\text{Q}\cdot\mathscr{O} \right\rangle = 0. \tag{5.4}$$

This identity also holds for gauge symmetries $\text{J} = \text{G}[\mathscr{X}]$ when supplemented by the corresponding BRST compensators $\mathscr{K}[\text{G}[\mathscr{X}]]$.

## 5.2 Bi-local total variations

The goal is to formulate a Slavnov–Taylor identity corresponding to the level-one Yangian generator $\widehat{\text{J}}$. However, we cannot proceed as above because it is not evident how to derive the equivalent of (5.1) for a bi-local generator consisting of two field variations ordered in a particular way involving the planar limit. Hence we need to find the equivalent of (5.1) for bi-local generators.

The central idea underlying the identity (5.1) is that the generator J is a variational operator which can act either on the operator $\mathscr{O}$ or on the action within the phase factor $\exp(i\mathscr{S})$. For some bi-local generator $\text{J}^a \wedge \text{J}^b$ we have two variations which should individually act either on the operator or on the phase factor. When both of them act on the same object $\mathscr{O}$, the result should be given by the bi-local action denoted by $(\text{J}^a \wedge \text{J}^b)\cdot\mathscr{O}$. Note that this makes proper sense only if the planar topology of $\mathscr{O}$ is a circle. When they act on two objects $\mathscr{O}_1$ and $\mathscr{O}_2$ which are disconnected in the planar topology, the result should be a product of the actions of the individual generators $\text{J}^a$ and $\text{J}^b$ which we shall denote by $(\text{J}^a\cdot\mathscr{O}_1)\wedge(\text{J}^b\cdot\mathscr{O}_2)$. This product of operators will somehow incorporate the planar ordering of the bi-local generator $\text{J}^a \wedge \text{J}^b$ when acting on two independent objects $\mathscr{O}_1$ and $\mathscr{O}_2$. It is not evident how to define this combination, not even whether there is a proper definition in the first place, so the notation $(\text{J}^a\cdot\mathscr{O}_1)\wedge(\text{J}^b\cdot\mathscr{O}_2)$ shall serve as a placeholder and we shall use it for book-keeping purposes only.[24]

---

[24]The idea behind book-keeping by means of a bi-local combination of the kind $\mathscr{O}_1 \wedge (\mathscr{O}_2 - \mathscr{O}_3)$ is that a certain contribution to $\mathscr{O}_2$ would always appear in the same place within a planar correlator as does a structurally equivalent contribution to $\mathscr{O}_3$. Now, if a symmetry implies that $\mathscr{O}_2 = \mathscr{O}_3$, the above bi-local combination, independently of how it may be defined, is zero due to linearity. If we can make all bi-local terms $\mathscr{O}_1 \wedge \mathscr{O}_2$ cancel, there will be no need for a precise definition.

More concretely, we propose the bi-local generalisation of (5.1) as

$$\big\langle (J^a \wedge J^b) \cdot \mathcal{O} + i(J^a \wedge J^b) \cdot \mathscr{S} \, \mathcal{O} \big\rangle$$
$$+ \big\langle i(J^a \cdot \mathscr{S}) \wedge (J^b \cdot \mathcal{O}) + i(J^a \cdot \mathcal{O}) \wedge (J^b \cdot \mathscr{S}) - (J^a \cdot \mathscr{S}) \wedge (J^b \cdot \mathscr{S}) \, \mathcal{O} \big\rangle = 0. \tag{5.5}$$

Here we suppose that $J^a \wedge J^b$ induces a total variation of the path integral,[25] so that the sum of all these variations is zero. Note that the last term originates from the constituents of $J^a \wedge J^b$ acting on different actions within the exponential factor $\exp(i\mathscr{S})$. The above conjecture is our best guess for a total variational identity involving bi-local generators. Conceivably, it is not entirely correct or correct only under certain conditions. In particular, we do not claim to understand how to define the terms on the second line. It would be very desirable to better understand and/or formally derive the above variational identity (5.5). For the time being, it will help us setting up Slavnov–Taylor identities for the bi-local generators. In the following Sec. 6 we will test the predicted equations for tree-level correlation functions involving up to four external fields in order to justify our proposal a posteriori.

## 5.3  Bi-local symmetries

Armed with the variational identities (5.1,5.5) for local and bi-local generators, we can now try construct identities for bi-local generators $J^a \wedge J^b$ to predict their action on some operator $\mathcal{O}$ within a correlator

$$\big\langle (J^a \wedge J^b) \cdot \mathcal{O} \big\rangle = \dots . \tag{5.6}$$

Here $\mathcal{O}$ must be some operator of circular topology in the planar sense, i.e. it must consist of a single trace of fields.

**Bi-local BRST symmetry.**   It will be easiest to start with the additional bi-local symmetries based on BRST transformations introduced in Sec. 4.3. This is because the constituent BRST transformations are exact symmetries which do not require compensating terms. We propose that the above form of Slavnov–Taylor identity (5.4) equivalently applies to the bi-local transformation $Q \otimes Q$

$$\big\langle (Q \otimes Q) \cdot \mathcal{O} \big\rangle - i \big\langle \mathscr{K}[Q \otimes Q] Q \cdot \mathcal{O} \big\rangle = 0. \tag{5.7}$$

This is consistent with the proposed variational identity (5.5) because all terms on the second line vanish due to the fact that the action is BRST-closed, $Q \cdot \mathscr{S} = 0$.[26] The remainder of the derivation works precisely as in the case of a generic local generator $J$, and it uses the symmetry statement (4.19) for the gauge-fixed action.

We will later test the identity for tree-level correlators involving up to four external fields, and find that it successfully predicts all the arising terms. However, it is conceivable that further terms are needed to accommodate for a larger number of fields in $\mathcal{O}$.

For the mixed bi-local generator $Q \wedge J$, we have to bear in mind that the constituent $J$ does not annihilate the action right away, but it requires a BRST compensator $\mathscr{K}[J]$. In order to cancel the arising terms, we will need a cascade of further compensating terms. Altogether we need to add up five variational identities (5.1) and (5.5) for various pairs of generators and

---

[25]It would be helpful to understand under which conditions on $J^a$ and $J^b$ this assumption holds and how far violations could break symmetries.

[26]Even though we do not understand the nature of such terms, we hope that we can apply symmetry statements to eliminate them.

operators to cancel all undesired terms,

$$
\begin{aligned}
0 = & \big\langle (Q \wedge J) \cdot \mathcal{O} + i(Q \wedge J) \cdot \mathscr{S} \, \mathcal{O} - i(J \cdot \mathscr{S}) \wedge (Q \cdot \mathcal{O}) \big\rangle \\
& + \big\langle i \mathscr{K}[J](Q \otimes Q) \cdot \mathcal{O} + i(Q \otimes Q) \cdot \mathscr{K}[J] \, \mathcal{O} - (Q \otimes Q) \cdot \mathscr{S} \, \mathscr{K}[J] \, \mathcal{O} - i(Q \cdot \mathscr{K}[J]) \wedge (Q \cdot \mathcal{O}) \big\rangle \\
& + \big\langle -iQ \cdot \mathscr{K}[Q \wedge J] \, \mathcal{O} - i \mathscr{K}[Q \wedge J] Q \cdot \mathcal{O} \big\rangle \\
& + \big\langle Q \cdot \mathscr{K}[J] \, \mathscr{K}[Q \otimes Q] \, \mathcal{O} - \mathscr{K}[J] Q \cdot \mathscr{K}[Q \otimes Q] \, \mathcal{O} + \mathscr{K}[J] \, \mathscr{K}[Q \otimes Q] Q \cdot \mathcal{O} \big\rangle \\
& + \big\langle J \cdot \mathscr{S} \, \mathscr{K}[Q \otimes Q] \, \mathcal{O} - iJ \cdot \mathscr{K}[Q \otimes Q] \, \mathcal{O} - i \mathscr{K}[Q \otimes Q] J \cdot \mathcal{O} \big\rangle \\
= & \big\langle (Q \wedge J) \cdot \mathcal{O} + i \mathscr{K}[J](Q \otimes Q) \cdot \mathcal{O} \big\rangle \\
& + \big\langle -i \mathscr{K}[Q \wedge J] Q \cdot \mathcal{O} + \mathscr{K}[J] \, \mathscr{K}[Q \otimes Q] Q \cdot \mathcal{O} - i \mathscr{K}[Q \otimes Q] J \cdot \mathcal{O} \big\rangle \\
& + i \big\langle \big( (Q \wedge J) \cdot \mathscr{S} - Q \cdot \mathscr{K}[Q \wedge J] + (Q \otimes Q) \cdot \mathscr{K}[J] - J \cdot \mathscr{K}[Q \otimes Q] \big) \mathcal{O} \big\rangle \\
& - i \big\langle \big( J \cdot \mathscr{S} + Q \cdot \mathscr{K}[J] \big) \wedge (Q \cdot \mathcal{O}) \big\rangle \\
& - \big\langle \big( (Q \otimes Q) \cdot \mathscr{S} + Q \cdot \mathscr{K}[Q \otimes Q] \big) \mathscr{K}[J] \, \mathcal{O} \big\rangle \\
& + \big\langle \big( J \cdot \mathscr{S} + Q \cdot \mathscr{K}[J] \big) \mathscr{K}[Q \otimes Q] \, \mathcal{O} \big\rangle .
\end{aligned}
\tag{5.8}
$$

Here we have immediately dropped terms $Q \cdot \mathscr{S}$ which vanish exactly by BRST symmetry. The last four lines after rearrangements of terms all vanish by level-zero and bi-local BRST symmetries of the action. The first of these is the symmetry statement (4.23) for the gauge-fixed action, and the desirable cancellation of structurally similar terms in the above formula serves as another justification for the extra terms (4.23). We are thus left with a Slavnov–Taylor identity for mixed bi-local BRST symmetries,

$$
\begin{aligned}
& \big\langle (Q \wedge J) \cdot \mathcal{O} + i \mathscr{K}[J](Q \otimes Q) \cdot \mathcal{O} \big\rangle \\
& \quad + \big\langle \big( -i \mathscr{K}[Q \wedge J] - \mathscr{K}[Q \otimes Q] \mathscr{K}[J] \big) Q \cdot \mathcal{O} - i \mathscr{K}[Q \otimes Q] J \cdot \mathcal{O} \big\rangle = 0 .
\end{aligned}
\tag{5.9}
$$

So we see that we need a number of terms to cancel off the effects of the gauge-fixing terms. However, it can be argued that all these terms are based on simpler symmetries acting on the operator $\mathcal{O}$. Importantly, all bi-local generators act on the operator $\mathcal{O}$ itself or are used for the BRST compensators. None of the bi-local combinations of operators $\mathcal{O}_1 \wedge \mathcal{O}_2$ remain, so there is no need to define them in practice because we can evaluate the Slavnov–Taylor identity for any single-trace operator $\mathcal{O}$. In App. A.3 we will demonstrate in an example that the above identity holds indeed.

**Level-one Yangian symmetry.** We are now well-prepared to compose the Slavnov–Taylor identity for the level-one Yangian generators $\widehat{J}$. It is obtained by iteratively cancelling terms arising due to partial integration via (5.1) and (5.5). We find that we can cancel all terms of the kind $\mathcal{O}_1 \wedge \mathcal{O}_2$ and all terms with a bare $\mathcal{O}$ not acted upon by some symmetry. The remaining terms form the Slavnov–Taylor identity[27]

$$
\begin{aligned}
& \big\langle \widehat{J} \cdot \mathcal{O} - i \mathscr{K}[J^{(1)}](Q \wedge J^{(2)}) \cdot \mathcal{O} - \tfrac{1}{2} \mathscr{K}[J^{(2)}] \mathscr{K}[J^{(1)}](Q \otimes Q) \cdot \mathcal{O} \big\rangle \\
& \quad + \big\langle \big( -i \mathscr{K}[\widehat{J}] - \mathscr{K}[Q \wedge J^{(2)}] \mathscr{K}[J^{(1)}] + \tfrac{i}{2} \mathscr{K}[Q \otimes Q] \mathscr{K}[J^{(2)}] \mathscr{K}[J^{(1)}] \big) Q \cdot \mathcal{O} \big\rangle \\
& \quad + \big\langle \big( -i \mathscr{K}[Q \wedge J^{(2)}] - \mathscr{K}[Q \otimes Q] \mathscr{K}[J^{(2)}] \big) J^{(1)} \cdot \mathcal{O} \big\rangle = 0 .
\end{aligned}
\tag{5.10}
$$

Again, a large number of extra terms are needed to compensate for the effects of gauge fixing. However, they are all based on simpler types of symmetries (bi-local BRST transformation or local generators).

---

[27]Note that $\mathscr{K}[J]$ is fermionic, so the combination $\mathscr{K}[J^{(1)}] \mathscr{K}[J^{(2)}]$ is non-zero despite its implicit anti-symmetry in the Sweedler notation for the generators.

In conclusion, we have proposed Slavnov–Taylor identities corresponding to all bi-local generators. These identities are all conjectural and they should be tested thoroughly until a proper justification for (5.5) can be found. We now proceed to test them by applying them to some elementary correlation functions.

# 6 Correlation functions

For correlation functions, symmetries manifest as Ward–Takahashi identities. In the gauge-fixed quantum theory, they follow by applying the Slavnov–Taylor identities to operators $\mathcal{O}$ consisting of a number of fields $Z_k(x_k)$ at different spacetime points. Our goal in this section is mainly to check whether the predictions due to the proposed Slavnov–Taylor identity are correct. With some gained confidence, one could later use the Ward–Takahashi identities towards constraining correlation functions in integrable planar gauge theories.

We would like to point out that analogous invariance results have been obtained in the context of scattering amplitudes starting with [10, 11]. Apart from the different kind of considered observables, the main difference between our analysis and previous studies lies in the underlying methods. Previous studies relied on existing expressions for scattering amplitudes or argued that constructions of scattering amplitudes preserve Yangian (dual conformal) symmetry. Further investigations used representations of scattering amplitudes in terms of advanced geometric concepts such as twistors, Graßmannians, etc. [13–16]. In our analysis we merely rely on invariance of the action and the structure of planar Feynman diagrams.

Note that analogous conclusions on Yangian invariance of planar correlation functions have been drawn in [21] within an integrable bi-scalar theory [22]. The main difference w.r.t. this analysis is that the bi-scalar theory is not a gauge theory and the representation of level-zero (conformal) symmetry is purely linear. Therefore almost all of the complications related to planar gauge theories that we shall encounter do not apply in the simplified bi-scalar field theory model.

## 6.1 Propagators

The simplest non-trivial correlation functions involve two fields; at tree level they are the propagators. It is essential to understand the Ward–Takahashi identities corresponding to both, local and bi-local generators because they will be needed for showing the identities for correlators with more than two legs.

**Local symmetries.** A first identity relating the ghost and auxiliary propagators comes from the Slavnov–Taylor identity (5.2) for BRST symmetry. We apply it to two fields $\mathcal{O} = Z_1 Z_2$ at tree level, and thus restrict to linear contributions to obtain

$$\left\langle Q_{[0]}{\cdot}Z_1\, Z_2 \right\rangle + \left\langle Z_1\, Q_{[0]}{\cdot}Z_2 \right\rangle = 0. \tag{6.1}$$

Now the only linear contributions to BRST variations are

$$Q_{[0]}{\cdot}A_\mu = \partial_\mu C \qquad \text{and} \qquad Q_{[0]}{\cdot}\bar{C} = B. \tag{6.2}$$

Considering that a correlator can be non-zero only if the overall ghost number is zero, a non-trivial statement follows only for $\mathcal{O} = \bar{C}(x)A_\mu(y)$. It amounts to

$$\left\langle B(x)A_\mu(y) \right\rangle = \left\langle \bar{C}(x)\,\partial_\mu C(y) \right\rangle = \frac{\partial}{\partial y^\mu}\left\langle \bar{C}(x)C(y) \right\rangle. \tag{6.3}$$

Apart from relating the auxiliary to the ghost propagator, the relationship states that the auxiliary field contracts only to the gauge degrees of freedom within the gauge field $A_\mu$. A simple corollary is that $\langle B F_{\mu\nu}\rangle = 0$ at tree level due to incompatible symmetrisations of the spacetime indices.

For a level-zero symmetry J we apply the Slavnov–Taylor identity (5.4) to two fields $\mathcal{O} = Z_1 Z_2$ and expand to tree level

$$\langle J_{[0]} {\cdot} Z_1 Z_2\rangle + \langle Z_1 J_{[0]} {\cdot} Z_2\rangle = i\big\langle \tfrac{1}{2}\mathcal{K}[J]_{[2]} Q_{[0]} {\cdot} Z_1 Z_2\big\rangle + i\big\langle \tfrac{1}{2}\mathcal{K}[J]_{[2]} Z_1 Q_{[0]} {\cdot} Z_2\big\rangle. \qquad (6.4)$$

The l.h.s. of this equation is the symmetry variation of a propagator. The equation shows that the propagator respects the symmetry manifestly up to terms involving linear BRST variations of the fields. One can convince oneself that all terms in (6.4) are trivially zero when one of the fields is $\bar{C}$.[28] A non-vanishing r.h.s. for (6.4) thus requires one of the fields to be a gauge potential. Explicitly, if the other field is a gauge potential as well, we find with (4.12)

$$\langle J_{[0]} {\cdot} A_\mu(x) A_\nu(y)\rangle + \langle A_\mu(x) J_{[0]} {\cdot} A_\nu(y)\rangle$$
$$= i\frac{\partial}{\partial x^\mu}\int dz^d \langle C(x)\partial^\rho \bar{C}(z)\rangle \langle J_{[0]} {\cdot} A_\rho(z) A_\nu(y)\rangle \qquad (6.5)$$
$$+ i\frac{\partial}{\partial y^\nu}\int dz^d \langle C(y)\partial^\rho \bar{C}(z)\rangle \langle A_\mu(x) J_{[0]} {\cdot} A_\rho(z)\rangle.$$

and otherwise for a generic gauge-covariant field $Z$

$$\langle J_{[0]} {\cdot} A_\mu(x) Z(y)\rangle + \langle A_\mu(x) J_{[0]} {\cdot} Z(y)\rangle = i\frac{\partial}{\partial x^\mu}\int dz^d \langle C(x)\partial^\rho \bar{C}(z)\rangle \langle J_{[0]} {\cdot} A_\rho(z) Z(y)\rangle. \qquad (6.6)$$

We note that the extra terms are total derivatives and as such constitute gauge variations.

**Bi-local symmetries.** For the corresponding level-one Ward–Takahashi identity we recall from [9] that level-one invariance of the quadratic part of the action follows from the corresponding level-zero invariance together with the vanishing of the dual Coxeter number of the level-zero symmetries. Indeed, we can derive a relationship from the level-zero identity (5.4) for the generator $J = J^{(1)}$ and the operator $\mathcal{O} = A_\mu(x) J^{(2)} {\cdot} A_\nu(y)$. Since $Z = J_{[0]} {\cdot} A_\nu$ is gauge covariant, we can read off the result from (6.6) as

$$\langle \widehat{J} {\cdot} \big(A_\mu(x) A_\nu(y)\big)\rangle = \langle J^{(1)}_{[0]} {\cdot} A_\mu(x) J^{(2)}_{[0]} {\cdot} A_\nu(y)\rangle$$
$$= i\frac{\partial}{\partial x^\mu}\int dz^d \langle C(x)\partial^\rho \bar{C}(z)\rangle \langle J^{(1)}_{[0]} {\cdot} A_\rho(z) J^{(2)}_{[0]} {\cdot} A_\nu(y)\rangle \qquad (6.7)$$
$$= i\frac{\partial}{\partial y^\nu}\int dz^d \langle C(y)\partial^\rho \bar{C}(z)\rangle \langle J^{(1)}_{[0]} {\cdot} A_\mu(x) J^{(2)}_{[0]} {\cdot} A_\rho(z)\rangle.$$

The second form follows in a similar fashion by exchanging the role of the two fields. Together the two identities imply that the correlator is a total derivative in both points

$$\langle J^{(1)}_{[0]} {\cdot} A_\mu(x) J^{(2)}_{[0]} {\cdot} A_\nu(y)\rangle = -\frac{\partial}{\partial x^\mu}\frac{\partial}{\partial y^\nu}, R[\widehat{J}](x, y), \qquad (6.8)$$

with the remainder function $R[\widehat{J}]$ taking the form

$$R[\widehat{J}](x, y) = \int dz_1^d\, dz_2^d \langle J^{(1)}_{[0]} {\cdot} A_\rho(z_1) J^{(2)}_{[0]} {\cdot} A_\sigma(z_2)\rangle \langle C(x)\partial^\rho \bar{C}(z_1)\rangle \langle C(y)\partial^\sigma \bar{C}(z_2)\rangle. \qquad (6.9)$$

---

[28]The generator $J_{[0]}$ never produces the ghost $C$, so $\bar{C}$ cannot contract with any other field. Furthermore, $J_{[0]}$ produces a gauge-covariant field, so it will not contract with $B \sim Q_{[0]} {\cdot} \bar{C}$.

This function has been evaluated explicitly for $\widehat{J} = \widehat{P}$ in $\mathcal{N} = 4$ sYM in [23] with a very simple structure as $R[\widehat{P}_\mu] \sim (x - y)_\mu/(x - y)^2$.

Let us point out one curiosity regarding the function $R$: On the one hand, the expression (6.9) heavily depends on ghost field propagators and therefore manifestly depends on the gauge-fixing procedure. On the other hand, the function must be independent of the chosen gauge because the underlying correlator on the l.h.s. of (6.8) involves gauge-covariant fields only.

Considering more general gauge theories, we can speculate that the function $R$ may be even more universal. This is based on the fact that gauge fields have mass dimension 1 and in (6.8) their vector index is used up for formulating the total derivatives. Consequently, the function $R[\widehat{J}]$ has the same mass dimension as the level-one generator $\widehat{J}$. If we furthermore expect it transform analogously to $\widehat{J}$, the only reasonable form appears to be

$$R[\widehat{J}] \sim (J_x - J_y) \log(x - y)^2, \tag{6.10}$$

where $J_x$ is the level-zero generator corresponding to $\widehat{J}$ acting on a spacetime point $x$ (rather than on fields), see also [24]. In the following we will not need the precise form for $R[\widehat{J}]$, but it would be desirable to understand better the above form (6.10) and whether there is a manifestly gauge-invariant way to derive it.

Let us finish by checking the bi-local Slavnov–Takahashi identity (5.10) applied to two fields $A_\mu(x)A_\nu(y)$ at tree level. Note that all compensators for bi-local operators are cubic at least, and consequently do not contribute at this level. For the remaining terms we find

$$\left\langle \widehat{J}_{[0]} \cdot \left( A_\mu(x) A_\nu(y) \right) \right\rangle = -\frac{\partial}{\partial x^\mu} \frac{\partial}{\partial y^\nu} R[\widehat{J}](x, y),$$

$$-i\left\langle \tfrac{1}{2}\mathcal{K}[J^{(1)}]_{[2]} (Q \wedge J^{(2)})_{[0]} \cdot \left( A_\mu(x) A_\nu(y) \right) \right\rangle = 2\frac{\partial}{\partial x^\mu} \frac{\partial}{\partial y^\nu} R[\widehat{J}](x, y), \tag{6.11}$$

$$\tfrac{1}{2}\left\langle \tfrac{1}{2}\mathcal{K}[J^{(2)}]_{[2]} \tfrac{1}{2}\mathcal{K}[J^{(1)}]_{[2]} (Q \otimes Q)_{[0]} \cdot \left( A_\mu(x) A_\nu(y) \right) \right\rangle = -\frac{\partial}{\partial x^\mu} \frac{\partial}{\partial y^\nu} R[\widehat{J}](x, y).$$

Indeed, they sum to zero, which serves as a first confirmation for the identity (5.10).

## 6.2 Level-zero symmetries of three-point functions

Next, we will discuss Ward–Takahashi identities for planar correlators of three fields. As before, we start with a level-zero generator $J$ to gain some experience in the occurring structures and required transformations. The Slavnov–Takahashi identity (5.4) for three fields reads

$$\left\langle J \cdot (Z_1 Z_2 Z_3) \right\rangle - i\left\langle \mathcal{K}[J] Q \cdot (Z_1 Z_2 Z_3) \right\rangle = 0. \tag{6.12}$$

When expanding out in the number of fields, we find the following contributions

$$\begin{aligned}
0 = & \left\langle J_{[0]} \cdot (Z_1 Z_2 Z_3) \right\rangle_{[1]} + \left\langle J_{[1]} \cdot (Z_1 Z_2 Z_3) \right\rangle_{[0]} \\
& - i\left\langle \tfrac{1}{2}\mathcal{K}[J]_{[2]} Q_{[0]} \cdot (Z_1 Z_2 Z_3) \right\rangle_{[1]} - i\left\langle \tfrac{1}{3}\mathcal{K}[J]_{[3]} Q_{[0]} \cdot (Z_1 Z_2 Z_3) \right\rangle_{[0]} \\
& - i\left\langle \tfrac{1}{2}\mathcal{K}[J]_{[2]} Q_{[1]} \cdot (Z_1 Z_2 Z_3) \right\rangle_{[0]},
\end{aligned} \tag{6.13}$$

where $\langle \dots \rangle_{[n]}$ represents a correlator with $n$ three-vertices inserted (we may suppress this notation for correlators without vertices). The former two terms represent the ordinary (non-linear) variation of the three-point function, while the latter three compensate for effects due to gauge fixing. As before, terms involving $Q_{[0]}$ produce total derivatives when acting on external gauge fields and nothing otherwise. Terms involving $Q_{[1]}$ yield a new kind of contribution

which is not a total derivative, but which is implied by proper non-linear gauge transformations.

By formally writing this expression in terms of propagators and vertices and by using the symmetry of propagators derived in Sec. 6.1, one can rewrite the above expression as

$$
\begin{aligned}
0 = & -i\left\langle \tfrac{1}{3}\mathrm{J}_{[0]}\!\cdot\!\mathscr{S}_{[3]}\, Z_1 Z_2 Z_3\right\rangle - i\left\langle \tfrac{1}{2}\mathrm{J}_{[1]}\!\cdot\!\mathscr{S}_{[2]}\, Z_1 Z_2 Z_3\right\rangle \\
& -i\left\langle \tfrac{1}{3}\mathrm{Q}_{[0]}\!\cdot\!\mathscr{K}[\mathrm{J}]_{[3]}\, Z_1 Z_2 Z_3\right\rangle - i\left\langle \tfrac{1}{2}\mathrm{Q}_{[1]}\!\cdot\!\mathscr{K}[\mathrm{J}]_{[2]}\, Z_1 Z_2 Z_3\right\rangle \\
& +\left\langle \tfrac{1}{3}\mathrm{Q}_{[0]}\!\cdot\!\mathscr{S}_{[3]}\,\tfrac{1}{2}\mathscr{K}[\mathrm{J}]_{[2]}\, Z_1 Z_2 Z_3\right\rangle + \left\langle \tfrac{1}{2}\mathrm{Q}_{[1]}\!\cdot\!\mathscr{S}_{[2]}\,\tfrac{1}{2}\mathscr{K}[\mathrm{J}]_{[2]}\, Z_1 Z_2 Z_3\right\rangle.
\end{aligned}
\tag{6.14}
$$

These terms are easily combined to

$$
-i\left\langle \tfrac{1}{3}(\mathrm{J}\!\cdot\!\mathscr{S} + \mathrm{Q}\!\cdot\!\mathscr{K}[\mathrm{J}])_{[3]}\, Z_1 Z_2 Z_3\right\rangle + \left\langle \tfrac{1}{3}(\mathrm{Q}\!\cdot\!\mathscr{S})_{[3]}\,\tfrac{1}{2}\mathscr{K}[\mathrm{J}]_{[2]}\, Z_1 Z_2 Z_3\right\rangle,
\tag{6.15}
$$

both of which are zero due to level-zero and BRST symmetry of the action, respectively, see Sec. 4.2. This confirms that the above Ward–Takahashi identity (6.12) holds at tree level by means of elementary transformations and symmetries of the action.

Unfortunately, the above transformation requires a lot of patience and care. Let us therefore present some useful identities and walk through a simplified transformation where we ignore all terms due to gauge fixing. We will also introduce a diagrammatical representation for the arising terms which helps to visually understand the conformation of the identity.

Some useful identities in transforming the expressions are as follows: First, the insertion of the quadratic part of the action between two propagators cancels one of the propagators

$$
\left\langle Z_1 \mathscr{S}_{[2],1}\right\rangle\left\langle Z_2 \mathscr{S}_{[2],2}\right\rangle = i\left\langle Z_1 Z_2\right\rangle.
\tag{6.16}
$$

Here we use a shorthand notation $\mathscr{X}_k$ for the $k$-th field within a polynomial $\mathscr{X}$ with an implicit sum over all monomials contributing to $\mathscr{X}$. In particular, the notation allows us to write the quadratic part of the action as

$$
\mathscr{S}_{[2],1}\mathscr{S}_{[2],2} := \sum_{\text{monomials}} \mathscr{S}_{[2],1}\mathscr{S}_{[2],2} = \mathscr{S}_{[2]},
\tag{6.17}
$$

with some similarity to Sweedler's notation introduced in (2.15). A diagrammatical representation of the above identity is simply

$$
\underline{\quad}\!\!\bigcirc\!\!\underline{\quad} = i \,\underline{\qquad}.
\tag{6.18}
$$

The lines correspond to propagators and the disc to the quadratic part of the action.

Second, a correlator of three fields at tree level requires the insertion of a cubic vertex

$$
\left\langle Z_1 Z_2 Z_3\right\rangle_{[1]} = i\left\langle Z_1 \mathscr{S}_{[3],2}\right\rangle_{[0]}\left\langle Z_2 \mathscr{S}_{[3],1}\right\rangle_{[0]}\left\langle Z_3 \mathscr{S}_{[3],3}\right\rangle_{[0]} = i
\tag{6.19}
$$

The central vertex in the diagram corresponds to the cubic part $\mathscr{S}_{[3]}$ of the action. Note that its symmetry factor $^1/_3$ is compensated by three sets of contractions which are equivalent due to the cyclic symmetry of the action. Here we have restricted to planar contributions which allows contractions between the external fields and the fields of the vertex in opposite ordering.

Third, the non-linear contribution of a symmetry generator on an external field at tree level requires no further vertex and it splits up into two propagators

$$\langle J_{[1]}{\cdot}Z_1\,Z_2 Z_3\rangle = \langle (J_{[1]}{\cdot}Z_1)_2\,Z_2\rangle\langle (J_{[1]}{\cdot}Z_1)_1\,Z_3\rangle = \quad. \tag{6.20}$$

The (blue) semi-circle in the diagram represents the level-zero symmetry generator J. It acts on the (single) field which in on the straight side, and yields (one or several) fields on the circular side.

Using the above rules, the symmetry variation of the three-field correlator can be transformed step-by-step as follows (we suppress all gauge-fixing terms; a complete derivation can be found in App. A.2):[29]

$$
\begin{aligned}
\langle J{\cdot}(Z_1 Z_2 Z_3)\rangle &\simeq 3\langle J_{[0]}{\cdot}Z_1 Z_2 Z_3\rangle + 3\langle J_{[1]}{\cdot}Z_1 Z_2 Z_3\rangle\\
&\simeq 3i\langle J_{[0]}{\cdot}Z_1 \mathscr{S}_{[3],2}\rangle\langle Z_2 \mathscr{S}_{[3],1}\rangle\langle Z_3 \mathscr{S}_{[3],3}\rangle + 3\langle (J_{[1]}{\cdot}Z_1)_2 Z_2\rangle\langle (J_{[1]}{\cdot}Z_1)_1 Z_3\rangle\\
&\simeq -3i\langle Z_1 J_{[0]}{\cdot}\mathscr{S}_{[3],2}\rangle\langle Z_2 \mathscr{S}_{[3],1}\rangle\langle Z_3 \mathscr{S}_{[3],3}\rangle\\
&\quad -3i\langle \mathscr{S}_{[2],1} Z_1\rangle\langle (J_{[1]}{\cdot}\mathscr{S}_{[2],2})_2 Z_2\rangle\langle (J_{[1]}{\cdot}\mathscr{S}_{[2],2})_1 Z_3\rangle\\
&\simeq -i\langle Z_1 Z_2 Z_3 \tfrac{1}{3}J_{[0]}{\cdot}\mathscr{S}_{[3]}\rangle - i\langle Z_1 Z_2 Z_3 \tfrac{1}{2}J_{[1]}{\cdot}\mathscr{S}_{[2]}\rangle\\
&= -i\langle Z_1 Z_2 Z_3 \tfrac{1}{3}(J{\cdot}\mathscr{S})_{[3]}\rangle = 0.
\end{aligned}
\tag{6.21}
$$

The sign '$\simeq$' here and in the following implies to equivalence modulo cyclic permutations of the three external fields. Using diagrams we can write the first class of terms as

$$\langle J_{[0]}{\cdot}(Z_1 Z_2 Z_3)\rangle = i \qquad + i \qquad + i$$

$$= -i \qquad - i \qquad - i \quad. \tag{6.22}$$

The transformation towards the second line makes use of the linearised symmetry of the propagators discussed in Sec. 6.1

$$\quad = -\quad. \tag{6.23}$$

The second class of terms corresponds to the diagrams

$$\langle J_{[1]}{\cdot}(Z_1 Z_2 Z_3)\rangle = \qquad + \qquad +$$

$$= -i \qquad - i \qquad - i \quad. \tag{6.24}$$

---

[29]The reduction of prefactors from 3 to $^1\!/_3$ and $^1\!/_2$ in the second but last line is due to two effects: $J_{[3-n]}{\cdot}\mathscr{S}_{[n]}$ produces $n$ equivalent terms and there are 3 equivalent contractions with the external fields up to cyclic permutations.

The diagrams on the second line are obtained by inserting the kinetic part of the action as in (6.18). Combining all terms, we find

$$\big\langle J_{[0]}\cdot(Z_1 Z_2 Z_3)\big\rangle_{[1]} + \big\langle J_{[1]}\cdot(Z_1 Z_2 Z_3)\big\rangle_{[0]} = -i \qquad \qquad \qquad \qquad \tag{6.25}$$

$$= -i\big\langle \tfrac{1}{3}(J\cdot\mathscr{S})_{[3]}\, Z_1 Z_2 Z_3\big\rangle_{[0]} = 0,$$

where the (green) vertex with embedded symmetry generator represents the variation of the action w.r.t. this symmetry. Such vertices represent a sum of terms which is zero by invariance of the action under this symmetry, $J\cdot\mathscr{S}=0$. This proves the level-zero Ward–Takahashi identity modulo gauge-fixing terms at the level of diagrams, see App. A.2 for a complete treatment.

### 6.3 Level-one symmetries of three-point functions

Now we are in a good position to discuss the level-one Ward–Takahashi identity. In order to streamline the calculation, we first define a manifestly cyclic variant $(J^a \wedge J^b)'$ of the bi-local generator $J^a \wedge J^b$ on the external fields $\mathscr{O}_{[n]} := Z_1 \cdots Z_n$ [30]

$$(J^a \wedge J^b)'\cdot\mathscr{O}_{[n]} := (J^a \wedge J^b)\cdot\mathscr{O}_{[n]} + \sum_{k=1}^{n}\frac{n+1-2k}{n}\big(J^a\cdot(J_k^b\mathscr{O}_{[n]}) - J^b\cdot(J_k^a\mathscr{O}_{[n]}) - [J_k^a, J_k^b]\mathscr{O}_{[n]}\big) \tag{6.26}$$

$$= \sum_{k=1}^{n}(J^a \wedge J^b)_k\mathscr{O}_{[n]} + \sum_{k=1}^{n}\sum_{j=1}^{n-1}\frac{2j-n}{n}J_{k+j}^a J_k^b \mathscr{O}_{[n]}.$$

It differs from the original definition by terms which contain the commutator of two local generators $[J^a, J^b] = 0$ as well as terms which are local transformations of some combinations of fields. It will be consistent to use the above definition of bi-local actions on fields within the Slavnov–Taylor identity (5.10). The additional terms cancel against each other upon use of the corresponding local identity (5.4).[31] Due to manifest cyclicity it now makes sense to compare modulo cyclic permutations

$$(J^{(1)} \otimes J^{(2)})'\cdot\mathscr{O}_{[n]} \simeq n(J^{(1)} \otimes J^{(2)})_1\mathscr{O}_{[n]} + \sum_{j=1}^{n-1}(j-\tfrac{1}{2}n)J_{j+1}^{(1)} J_1^{(2)}\mathscr{O}_{[n]}. \tag{6.27}$$

For three external fields this expression reduces to

$$\big\langle \widehat{J}'\cdot(Z_1 Z_2 Z_3)\big\rangle \simeq 3\big\langle Z_1 Z_2 \widehat{J}\cdot Z_3\big\rangle + \big\langle J^{(1)}\cdot Z_1\, J^{(2)}\cdot Z_2\, Z_3\big\rangle. \tag{6.28}$$

Upon evaluation of the correlators at tree level, we find the following diagrams

$$\big\langle \widehat{J}'\cdot(Z_1 Z_2 Z_3)\big\rangle \simeq 3 \qquad + i \qquad + \qquad + \qquad . \tag{6.29}$$

The double semi-circle represents the local contribution to the level-one generator $\widehat{J}$ (with no linear contribution), while the single semi-circles without and with decoration (dot) correspond to the level-zero generators $J^{(1)}$ and $J^{(2)}$, respectively.

---

[30] Here, $J_k$ is the generator $J$ acting on the field(s) at the $k$-th site of the correlator (rather than the $k$-th field with in the polynomial). It is the fully non-linear generator which may change the length of the polynomial.

[31] This perfect cancellation can be viewed as a mild verification of the form (5.10) of the bi-local Slavnov–Taylor identity.

By using the transformations introduced above and by discarding all gauge-fixing contributions we can make all the symmetry generators act on a central vertex

$$\widehat{\mathrm{J}}\langle Z_1 Z_2 Z_3\rangle \simeq -3i \quad + i \quad + i \quad - i$$

$$\simeq -i \quad = -i\big\langle \tfrac{1}{3}(\widehat{\mathrm{J}}{\cdot}\mathscr{S})_{[3]}\, Z_1 Z_2 Z_3\big\rangle. \tag{6.30}$$

The combination of insertions is proportional to the cubic combination of local terms which arises in Yangian invariance of the action (2.46),

$$(\widehat{\mathrm{J}}{\cdot}\mathscr{S})_{[3]} \simeq 3\widehat{\mathrm{J}}_{[1],1}\mathscr{S}_{[2]} + \mathrm{J}^{(2)}_{[0],2}\mathrm{J}^{(1)}_{[0],1}\mathscr{S}_{[3]} + \big(\mathrm{J}^{(2)}_{[0],1} - \mathrm{J}^{(2)}_{[0],2}\big)\mathrm{J}^{(1)}_{[1],1}\mathscr{S}_{[2]} \simeq 0. \tag{6.31}$$

Note that the natural ordering for the fields on the insertion is opposite to the ordering of external fields. In fact, this reversal of ordering is necessary to make the signs match up: Shifting a local term from the external legs to the internal vertices typically flips the sign. Shifting a bi-local term, however, induces one sign flip for each propagator. Hence, an extra sign flip is needed to match the sign of the transformation of local terms. It arises due to changing the order in combination with the anti-symmetry of the bi-local terms. The latter anti-symmetry also implies that the interchange of $\mathrm{J}^{(1)}$ and $\mathrm{J}^{(2)}$ is equivalent to a flip of sign.

In conclusion, the level-one Ward–Takahashi identity for correlators of three fields is equivalent to the cubic contribution to the level-one symmetry of the action (modulo gauge fixing). We will redo a similar calculation with all gauge-fixing terms in App. A.3.

Finally, we would like to point out the structural difference between the bi-local action on a collection of external fields (6.26) and the one on cyclic polynomials (2.46) such as the action $\mathscr{S}$. Both expressions have similar terms with similar coefficients, but their details differ. In particular, the former derives directly from the bi-local action on open polynomials (2.45), and consequently there are no overlapping terms. We have seen above that both types of expressions are indeed related by the Slavnov–Taylor identity (5.10), and it seems natural to apply these particular actions for each of the objects that are acted upon. Curiously, the conjectured bi-local variational identity (5.5) which was used to derive the identity (5.10) seems to implicitly translate between the two forms without making direct reference to either. It would be good to understand this issue better.[32]

## 6.4 Level-one symmetries of four-point functions

Finally, let us investigate the level-one Ward–Takahashi identity for four external fields. The new element of this calculation is that the four-point correlation function has an internal propagator. It will be interesting to see how the non-linear contributions conspire to relate the different topologies of Feynman graphs for this correlator, and it will be reassuring to see that the identity works out in this more elaborate case. For conciseness we will again drop all terms due to gauge fixing.

---

[32]The different forms of applicable bi-local actions may be related to the different quantum field theory functionals underlying the different objects: Correlation functions are based on the partition function (or its connected component in view of the planar limit) which is a functional of field sources. Conversely, the action (and likewise the effective action) is a functional of the fields themselves. It is conceivable that the Legendre transformation, which translates between these two types of objects, naturally maps one type of bi-local representation on cyclic objects to the other (while there is no qualitative change for local representations).

Again, we start with the level-one generator acting on four fields modulo cyclic permutations (6.27)

$$\langle \widehat{J}'\cdot(Z_1 Z_2 Z_3 Z_4)\rangle \simeq 4\langle \widehat{J}\cdot Z_1\, Z_2 Z_3 Z_4\rangle + 2\langle J^{(1)}\cdot Z_1\, J^{(2)}\cdot Z_2\, Z_3 Z_4\rangle. \tag{6.32}$$

Expanding in terms of planar diagrams, we find the following terms at tree level

Here we have used the level-zero symmetry of the propagator to bring the individual terms to some standard form which makes them easier to compare (see Sec. 6.2)

We add a carefully balanced sum of terms which all involve a composite cubic interaction vertex which is zero by means of level-zero or level-one symmetry (see Sec. 6.2 and 6.3) or the composite operator representing the commutator $J^{(1)} J^{(2)} = 0$

$$+ \frac{4i}{3} \enspace + \frac{4i}{3} \enspace + \frac{4i}{3} \enspace ,$$

$$+ \frac{i}{3} \enspace \simeq + \frac{i}{3} \enspace + \frac{i}{3} \enspace - \frac{i}{3} \enspace ,$$

$$- \frac{i}{3} \enspace \simeq - \frac{i}{3} \enspace - \frac{i}{3} \enspace + \frac{i}{3} \enspace . \tag{6.34}$$

In writing these terms, we have again made use of simple identities introduced in Sec. 6.2 and 6.3, such as level-zero symmetry of propagators, removal of quadratic vertices, vanishing of the dual Coxeter number, implicit anti-symmetry of the level-zero generators and cyclic permutations. By summing up all terms, we find that almost all diagrams cancel.

We arrive at a collection of terms with a central quartic vertex which match precisely with the quartic term in the invariance of the action (2.46),

$$
\begin{aligned}
(\widehat{J} \cdot \mathscr{S})_{[4]} &\simeq 4 \widehat{J}_{[1],1} \mathscr{S}_{[3]} + 2 J^{(2)}_{[0],2} J^{(1)}_{[0],1} \mathscr{S}_{[4]} + \big( J^{(2)}_{[1],1} - J^{(2)}_{[1],2} \big) J^{(1)}_{[1],1} \mathscr{S}_{[2]} \\
&\quad + \big( J^{(2)}_{[0],1} - J^{(2)}_{[0],2} + J^{(2)}_{[0],3} - J^{(2)}_{[0],4} \big) J^{(1)}_{[1],1} \mathscr{S}_{[3]} \simeq 0.
\end{aligned}
\tag{6.35}
$$

We can subtract these terms,

$$0 = i \enspace = i \Big\langle \tfrac{1}{4} (\widehat{J} \cdot \mathscr{S})_{[4]} Z_1 Z_2 Z_3 Z_4 \Big\rangle$$

$$\simeq 4i \enspace - 2i \enspace + \enspace - \enspace \tag{6.36}$$

$$+ i \enspace - i \enspace - i \enspace + i \enspace ,$$

and find that all terms cancel,

$$\big\langle \widehat{J}' \cdot (Z_1 Z_2 Z_3 Z_4) \big\rangle = 0. \tag{6.37}$$

This proves the level-one Ward–Takahashi identity for four external fields modulo gauge-fixing terms.

# 7 Yangian symmetry in the quantum theory

So far we have discussed Yangian symmetry for correlation functions only at tree level. Tree level is equivalent to the classical (non-linear) theory,[33] so we have established implications of Yangian symmetry for classical planar gauge theories based on [8,9]. It will thus be interesting to understand how Yangian symmetry extends to the quantum theory. Due to the various technical challenges, we shall only perform one test in a simplified setting and provide basic and qualitative arguments in this article.

---

[33] The generating functional of trees is the Legendre transform of the classical action.

## 7.1 Loop effects

At loop level many new issues may come into play: Proper treatment of loops in correlation functions will require renormalisation. Likewise, the symmetry generators need to be regularised and potentially renormalised. At the end of the day, there are three conceivable outcomes for the Slavnov–Taylor identities (5.10) corresponding to Yangian symmetries at loop level:

- Yangian symmetry is manifest: the tree-level form of the Slavnov–Taylor identities remains valid without changes at loop level.

- Yangian symmetry is anomalous: the Slavnov–Taylor identities receive anomalous contributions, but they constitute exact statements for the quantum theory.

- Yangian symmetry is broken: conceivably, quantum effects completely spoil the Slavnov–Taylor identities, and no meaningful conclusions due to Yangian symmetry can be drawn for quantum observables.

We can speculate that our identities will at least continue to hold as they stand at the level of loop integrands, i.e. before performing loop integrals. Divergences in the loop integrals and the ensuing renormalisation procedure in conjunction with gauge fixing and unphysical degrees of freedom could well render Yangian symmetry anomalous in some sense. Nevertheless it is also conceivable that the unusual kinds of transformations which came to use will not be compatible with loop integrands and thus spoil the identities beyond repair. However, the many successes of integrability in $\mathcal{N} = 4$ sYM, at the loop level and even at finite coupling strength, see [1], suggest that the identities will remain largely intact in the full quantum theory. Let us therefore inspect the simplest non-trivial variation of a correlation function at loop level.

## 7.2 Three-point function at one loop

The one-loop correction to the two-point function is the simplest quantum correction to a correlation function. However, we have argued in Sec. 6.1 that two-point functions are more or less trivially invariant under Yangian level-one symmetries (up to gauge fixing). We therefore consider the three-point function at one loop as the next-to-simplest correlator.

As can be seen in App. A, a proper treatment of gauge fixing bloats the combinatorics without altering the conclusions concerning invariance. We shall therefore ignore gauge-fixing effects in the following analysis. Furthermore, we will disregard regularisation and renormalisation which would be needed to properly eliminate divergences at loop level. Effectively, we will thus consider loop integrands rather than loop integrals. On the one hand, this restricts the significance of the result. On the other hand, we will demonstrate that the non-linear representation of Yangian symmetry is structurally compatible with loop-level planar diagrams. The many non-trivial cancellations in our example will make it appear likely that Yangian symmetry does not break in a bad way quantum mechanically. It would thus remain to argue against quantum anomalies which we shall do later in this section.

We start with the level-one symmetry variation of the three-point function at one loop in terms of the planar diagrams introduced in Sec. 6.2

$$\left\langle \widehat{J}' \cdot (Z_1 Z_2 Z_3) \right\rangle_{(1)}$$

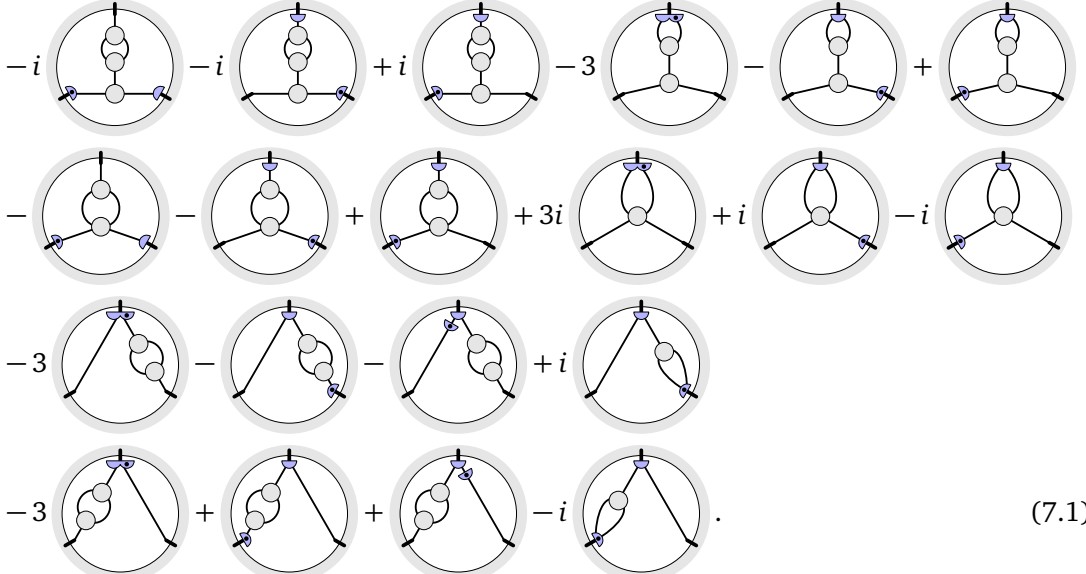

$$\tag{7.1}$$

We have already applied several identities to transform the diagrams to equivalent ones without pointing out these transformations explicitly: Quadratic vertices were eliminated by means of Green's identity (6.18). Level-zero invariance is used to shift linearised generators across propagators, see (6.23). Furthermore we consider diagrams modulo cyclic permutations, and we can exchange the two constituent level-zero generators (with or without dot) at the expense of a sign flip. Finally, we drop tadpole diagrams (any diagram where a propagator connects a point to itself).[34]

We need to show that all these diagrams sum up to zero. We achieve this by adding further diagrams which we already know to sum to zero due to invariance of the action under level-zero and level-one symmetries (green circles) as well as commutativity of the level-zero constituents of the level-one generators (green boxes):

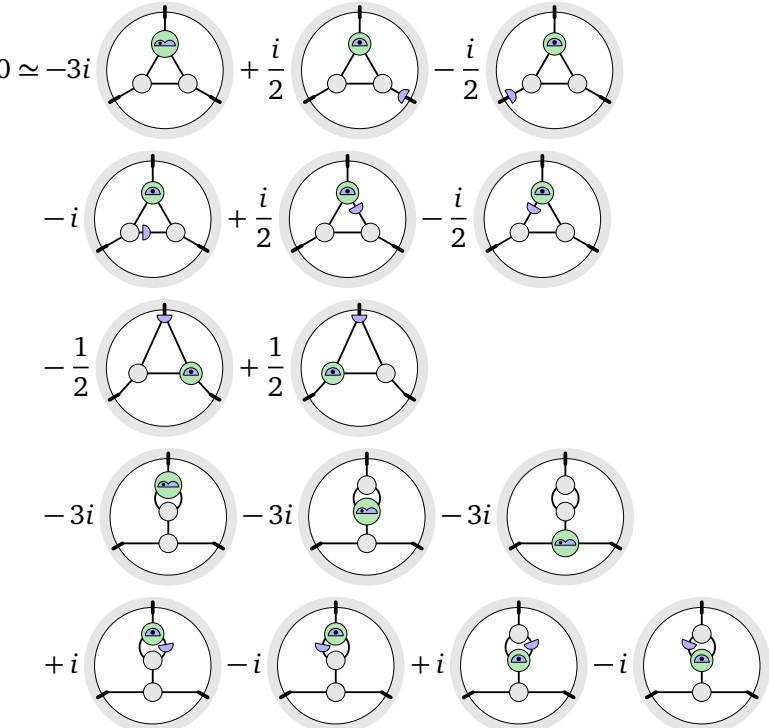

---

[34]The loop integral of this propagator merely amounts to some (potentially divergent) number, which is equivalent to a suitably chosen local counterterm and can therefore be removed.

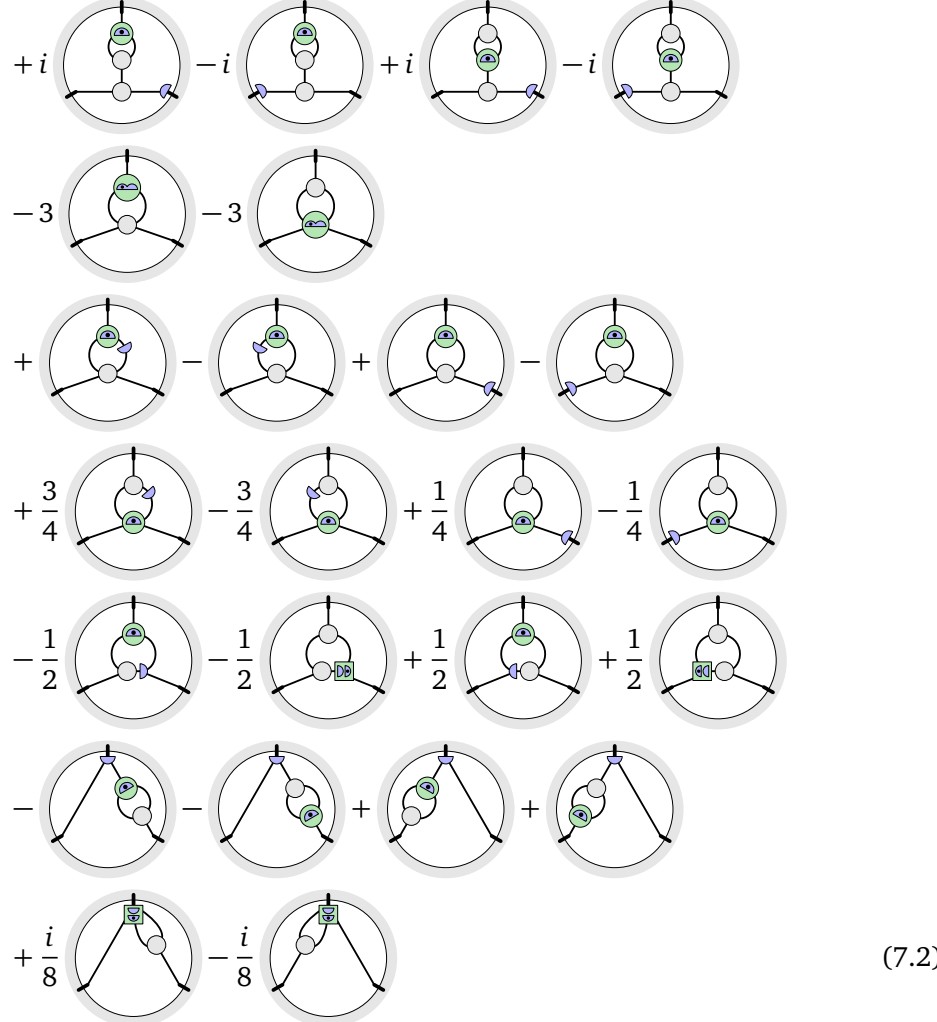

$$(7.2)$$

Each of these terms can be expanded in terms of elementary diagrams as before. They yield between 2 and 28 individual diagrams which can be transformed and brought to one of 90 standard forms using the transformation rules mentioned above, see App. B for a complete expansion of all terms. By careful inspection we indeed confirm that all the diagrams cancel

$$\left\langle \widehat{J}' \cdot (Z_1 Z_2 Z_3) \right\rangle_{(1)} = 0. \tag{7.3}$$

In our example, we find no surprises for Yangian symmetry of loop integrands nor does the invariance require modification of the symmetry representation. However, it remains to be seen whether gauge fixing or regularisation will change this conclusion.

## 7.3 Anomalies

Finally, we would like to comment on anomalies, see [25], in the interpretation as a non-invariance of the quantum mechanical path integral measure [26, 27]. For ordinary symmetries, the anomaly could be understood a deformation of the total variational identity (5.1)

$$\left\langle J \cdot \mathcal{O} + i J \cdot \mathcal{S} \, \mathcal{O} + i \mathcal{A}[J] \, \mathcal{O} \right\rangle = 0. \tag{7.4}$$

Here the additional term $\mathcal{A}[J]$ represents the potentially non-trivial variation of the path integral measure by the generator J. A corresponding deformation of the conjectured variational

identity (5.5) for a bi-local generator $J^a \wedge J^b$ reads

$$
\begin{aligned}
0 = & \left\langle (J^a \wedge J^b) \cdot \mathcal{O} + i (J^a \wedge J^b) \cdot \mathscr{S} \, \mathcal{O} + i \mathscr{A}[J^a \wedge J^b] \, \mathcal{O} \right\rangle \\
& + \left\langle i(J^a \cdot \mathscr{S}) \wedge (J^b \cdot \mathcal{O}) + i(J^a \cdot \mathcal{O}) \wedge (J^b \cdot \mathscr{S}) + i \mathscr{A}[J^a] \wedge (J^b \cdot \mathcal{O}) + i(J^a \cdot \mathcal{O}) \wedge \mathscr{A}[J^b] \right\rangle \\
& + \left\langle -(J^a \cdot \mathscr{S}) \wedge (J^b \cdot \mathscr{S}) \, \mathcal{O} - \mathscr{A}[J^a] \wedge (J^b \cdot \mathscr{S}) \, \mathcal{O} - (J^a \cdot \mathscr{S}) \wedge \mathscr{A}[J^b] \, \mathcal{O} - \mathscr{A}[J^a] \wedge \mathscr{A}[J^b] \, \mathcal{O} \right\rangle .
\end{aligned}
\tag{7.5}
$$

Now the level-zero symmetry is non-anomalous in the principal examples of Yangian symmetric planar gauge theories, $\mathscr{A}[J] = 0$. Dropping all terms in the bi-local variation containing the level-zero anomaly $\mathscr{A}[J]$, we are left with the genuine anomaly $\mathscr{A}[J^a \wedge J^b]$ of a bi-local symmetry. So the question of a Yangian anomaly translates to the presence of this latter term.

A key aspect of anomalies is locality and cohomology of the symmetry algebra: In a renormalisable quantum field theory, the anomaly term $\mathscr{A}[J]$ is local. Moreover, the anomaly typically joins the variation of the action in the combination $J \cdot \mathscr{S} + \mathscr{A}[J]$,

$$
\left\langle J \cdot \mathcal{O} + i \big( J \cdot \mathscr{S} + \mathscr{A}[J] \big) \, \mathcal{O} \right\rangle = 0.
\tag{7.6}
$$

So the actual question is whether this combination vanishes or can be made to vanish by adjusting the action appropriately. Now the action is manifestly local and so is its variation. However, a local anomaly is not necessarily the variation of a local term, which makes this question a problem of cohomology. For the bi-local Yangian symmetry we can collect the terms of the above variational identity as

$$
\begin{aligned}
0 = & \left\langle \widehat{J} \cdot \mathcal{O} \right\rangle + i \left\langle \big( \widehat{J} \cdot \mathscr{S} + \mathscr{A}[\widehat{J}] \big) \, \mathcal{O} \right\rangle \\
& + i \left\langle \big( J^{(1)} \cdot \mathscr{S} + \mathscr{A}[J^{(1)}] \big) \wedge \big( J^{(2)} \cdot \mathcal{O} \big) \right\rangle \\
& - \left\langle \big( J^{(1)} \cdot \mathscr{S} + \mathscr{A}[J^{(1)}] \big) \otimes \big( J^{(2)} \cdot \mathscr{S} + \mathscr{A}[J^{(2)}] \big) \, \mathcal{O} \right\rangle .
\end{aligned}
\tag{7.7}
$$

One should therefore ask whether $\widehat{J} \cdot \mathscr{S} + \mathscr{A}[\widehat{J}] = 0$ together with $J \cdot \mathscr{S} + \mathscr{A}[J] = 0$. Importantly, is $\mathscr{A}[\widehat{J}]$ local? Is the appropriate cohomology for the Yangian algebra trivial? Can one construct a local anomaly term $\mathscr{A}[\widehat{J}]$ consistent with the relations of Yangian symmetry? And even if so, is the resulting prefactor in $\mathscr{A}[\widehat{J}]$ zero? Answering these questions should help in settling Yangian symmetry as a symmetry of a planar quantum gauge theory.

## 7.4 Anomalies in $\mathcal{N} = 4$ sYM

Ignoring potential subtleties regarding regularisation, gauge fixing, bi-local total variations and locality of $\mathscr{A}[\widehat{J}]$, we can consider the anomaly of the level-one momentum generator $\widehat{P}$ in $\mathcal{N} = 4$ sYM theory. Such an anomaly should obey a consistency relation along the lines of [28]

$$
J \cdot \mathscr{A}[\widehat{P}] = \widehat{P} \cdot \mathscr{A}[J] + \mathscr{A}\big[ [J, \widehat{P}] \big],
\tag{7.8}
$$

with the level-zero generators $J$ originating from the algebra at level one. Following the arguments in Sec. 3.3 and generalising (3.18) the commutator $[J, \widehat{P}]$ should be a combination of level-one generators following from the Yangian algebra as well as bi-local and local gauge transformations,

$$
[J, \widehat{P}] = [J, \widehat{P}]_Y + G[P^{(1)} \cdot \ldots] \wedge P^{(2)} + G[\widehat{P} \cdot \ldots].
\tag{7.9}
$$

According to (3.20) one can expect the anomalies of bi-local gauge transformations to reduce to anomalies of level-zero symmetries and a term two level-zero generators

$$
\mathscr{A}\big[ G[P^{(1)} \cdot \ldots] \wedge P^{(2)} \big] \simeq 2 I[P^{(1)} \cdot \ldots] \cdot \mathscr{A}[P^{(2)}] + 2 \mathscr{A}\big[ I[P^{(2)} \cdot P^{(1)} \cdot \ldots] \big].
\tag{7.10}
$$

In $\mathcal{N} = 4$ sYM there are no anomalies for superconformal and gauge symmetries and moreover the combination $\mathrm{P}^{(2)}\mathrm{P}^{(1)}$ vanishes, see (2.49). Hence the above consistency requirement reduces to

$$\mathrm{J}\cdot\mathscr{A}[\widehat{\mathrm{P}}] = \mathscr{A}\big[[\mathrm{J},\widehat{\mathrm{P}}]_{\mathrm{Y}}\big], \tag{7.11}$$

where the commutator is evaluated in the plain Yangian algebra. Since the level-one generators transform in the adjoint representation of the level-zero algebra, $\widehat{\mathrm{P}}$ has (almost) the same quantum numbers as the momentum generator P. Effectively, this implies that $\mathscr{A}[\widehat{\mathrm{P}}]$ is a translation-invariant supersymmetric Lorentz-vector SU(4)-singlet gauge-invariant operator of dimension 1. Furthermore, we can argue that the anomaly term has negative SU($N_c$)-parity just as the level-one Yangian generators.[35] Translation-invariant operators are integrals over local operators, and supersymmetric operators reside at the top of supermultiplets. Before integration over 4-dimensional spacetime, we therefore consider Lorentz-vector SU(4)-singlet gauge-invariant local operators of dimension $5 = 4 + 1$ and negative SU($N_c$)-parity at the top of some supermultiplet.

Now it is not difficult to argue that no such local operators exist based on the representation theory of the superconformal algebra $\mathfrak{psu}(2,2|4)$, see [29]: Long supermultiplets span a range of conformal dimension $8 = 4\mathcal{N}/2$, hence supersymmetric local operators at dimension less than 10 could possibly originate only from short supermultiplets. The only short supermultiplets relevant in the planar limit are the $1/2$-BPS single-trace multiplets. Negative SU($N_c$)-parity restricts to the $1/2$-BPS multiplet at dimension 3 which possesses no suitable supersymmetric state either. For example, it is easy to see that the SU(4) charge of the corresponding superconformal primary operators cannot be eliminated by the application of merely 4 supersymmetry operators which are needed to go from the primary at dimension 3 to a descendant at dimension 5.

We can also argue less abstractly by enumerating local operators with the desired properties: Dropping the requirement of supersymmetry, there are 15 Lorentz-vector SU(4)-singlet gauge-invariant local operators of dimension 5 and negative SU($N_c$)-parity. Eight of these are ordinary local operators[36]

$$\begin{aligned}
&\mathrm{tr}\big[\{\Phi_m,\Phi_n\}[\nabla_\mu\Phi_m,\Phi_n]\big], && \Gamma_\mu^{\alpha\beta}\,\mathrm{tr}\big[[\bar{\Psi}_\alpha,\Psi_\beta]\Phi_m\Phi_m\big], \\
&\Gamma_{\mu[mn]}^{\alpha\beta}\,\mathrm{tr}\big[\{\bar{\Psi}_\alpha,\Psi_\beta\}\Phi_m\Phi_n\big], && \Gamma_{\mu[mn]}^{\alpha\beta}\,\mathrm{tr}\big[\bar{\Psi}_\alpha\Phi_m\Psi_\beta\Phi_n\big], \\
&\Gamma_\nu^{\alpha\beta}\,\mathrm{tr}\big[[\bar{\Psi}_\alpha,\Psi_\beta]F_\mu{}^\nu\big], && \Gamma_\nu^{\alpha\beta}\,\mathrm{tr}\big[[\bar{\Psi}_\alpha,\Psi_\beta]\tilde{F}_\mu{}^\nu\big], \\
&\Gamma_m^{\alpha\beta}\,\mathrm{tr}\big[[\nabla_\mu\Psi_\alpha,\Psi_\beta]\Phi_m\big], && \Gamma_m^{\alpha\beta}\,\mathrm{tr}\big[[\nabla_\mu\bar{\Psi}_\alpha,\bar{\Psi}_\beta]\Phi_m\big],
\end{aligned} \tag{7.12}$$

four are conformal descendants

$$\begin{aligned}
&\partial^\nu\,\mathrm{tr}\big[\{\Phi_m,\Phi_m\}F_{\mu\nu}\big], && \partial^\nu\,\mathrm{tr}\big[\{\Phi_m,\Phi_m\}\tilde{F}_{\mu\nu}\big], \\
&\Gamma_{m\mu\nu}^{\alpha\beta}\partial^\nu\,\mathrm{tr}\big[[\Psi_\alpha,\Psi_\beta]\Phi_m\big], && \Gamma_{m\mu\nu}^{\alpha\beta}\partial^\nu\,\mathrm{tr}\big[[\bar{\Psi}_\alpha,\bar{\Psi}_\beta]\Phi_m\big],
\end{aligned} \tag{7.13}$$

and three are trivial modulo the equations of motion[37]

$$\begin{aligned}
&\mathrm{tr}\big[\Phi_m\Phi_m\nabla^\nu F_{\mu\nu}\big] + \dots, \\
&(\Gamma_\nu\Gamma_{m\mu})^{\alpha\beta}\,\mathrm{tr}\big[[\nabla^\nu\Psi_\alpha,\Psi_\beta]\Phi_m\big] + \dots, \\
&(\Gamma_\nu\Gamma_{m\mu})^{\alpha\beta}\,\mathrm{tr}\big[[\nabla^\nu\bar{\Psi}_\alpha,\bar{\Psi}_\beta]\Phi_m\big] + \dots.
\end{aligned} \tag{7.14}$$

---

[35]The SU($N_c$)-parity operation is the $\mathbb{Z}_2$ outer automorphism of SU($N_c$) which maps the adjoint fields $Z$ to their negative transpose $-Z^\mathsf{T}$. Therefore it effectively reverses the order of the trace in the planar limit. The antisymmetric combination of the bi-local generators implies a negative parity.

[36]We merely sketch their form using the fields of $\mathcal{N} = 4$ sYM in the following convention: $\Phi_m$ denotes the 6 scalar fields while $\Psi_\alpha,\bar{\Psi}_\alpha$ denote the spinor fields with $\alpha$ a combined spinor index to be acted upon by gamma-matrices $\Gamma_\mu, \Gamma_m$ in $4+6$ dimensions.

[37]It suffices to state the leading term with the largest number of derivatives as the others are linear combinations of the previously mentioned terms.

All of these belong to long superconformal multiplets: There are $1, 4, 1$ relevant long superconformal multiplets of primary dimension $3, 4, 5$, respectively [30]. Furthermore, there are 2 and 1 extra superconformal multiplets of primary dimension 4.5 and 5, respectively, which are proportional to the equations of motion. In the following table, we list the quantum numbers of the superconformal primaries in terms of conformal dimension $\Delta$ and Dynkin labels of the stability subgroups $SL(2, \mathbb{C})$ and $SU(4)$ of the superconformal primary condition. We furthermore sketch the combination of supercharges $Q, \bar{Q}$ and momentum generators $P$ that turns the superconformal primaries into descendant operators of the desired type ($\Delta = 5$, $[1, 1]$, $[0, 0, 0]$):

| $\Delta$ | $SL(2, \mathbb{C})$ | $SU(4)$ | descendant |
|---|---|---|---|
| 3 | $[0, 0]$ | $[0, 1, 0]$ | $Q\bar{Q}^3, Q^3\bar{Q}, Q^2P, \bar{Q}^2P$ |
| 4 | $[0, 0]$ | $[1, 0, 1]$ | $Q\bar{Q}$ |
| 4 | $[2, 0]$ | $[0, 0, 0]$ | $Q\bar{Q}, P$ |
| 4 | $[0, 2]$ | $[0, 0, 0]$ | $Q\bar{Q}, P$ |
| 4 | $[1, 1]$ | $[0, 1, 0]$ | $Q^2, \bar{Q}^2$ |
| 5 | $[1, 1]$ | $[0, 0, 0]$ | 1 |
| 4.5 | $[1, 0]$ | $[1, 0, 0]$ | $\bar{Q}$ |
| 4.5 | $[0, 1]$ | $[0, 0, 1]$ | $Q$ |
| 5 | $[1, 1]$ | $[0, 0, 0]$ | 1 |

(7.15)

Altogether these account for 8 conformal primaries, 4 conformal descendants ($P$) and 3 operators proportional to the equations of motions. As members of long supermultiplets, they are clearly not the highest and thus not supersymmetric states.

Therefore, there are no suitable operators $\mathscr{A}[\widehat{P}]$ in planar $\mathcal{N} = 4$ sYM theory, and consequently, Yangian symmetry is non-anomalous (modulo subtleties alluded to above and supposing the superconformal and gauge symmetries are non-anomalous). In fact, the absence of suitable operators $\mathscr{A}[\widehat{P}]$ not only implies the absence of anomalies, but even better, it implies (without subtleties) that the classical action must be invariant, as was shown explicitly in [9]. This constitutes an alternative proof of Yangian symmetry of planar $\mathcal{N} = 4$ sYM which is merely based on the representation content of the theory together with the Yangian algebra relations.

By the same logic, the level-one bonus symmetry $\widehat{B}$ [31–33] can be argued to be non-anomalous: The anomaly $\mathscr{A}[\widehat{B}]$ must be a translation-invariant Lorentz-singlet $SU(4)$-singlet gauge-invariant operator of dimension 0 and negative parity. Such an operator does not exist as one can show by straight-forward enumeration of local operators at dimension 4, see also (7.15). Even though this argument is much easier than the one for $\widehat{P}$, it may also be more fragile at the quantum level: Namely, the generator $\widehat{B}$ is based on superconformal boosts as opposed to $\widehat{P}$ which merely uses rotations, translations, supersymmetries and scale transformations. The issue is that special conformal symmetries are typically superficially broken by the quantisation procedure whereas rotations, translations and supersymmetries are known to hold manifestly. Therefore, an anomaly of $\widehat{P}$ is most closely linked to the anomaly of scale transformations, whereas the anomaly of $\widehat{B}$ requires understanding of the anomaly of special superconformal transformations.

## 8 Conclusions and Outlook

In this work, we have proposed and verified Ward–Takahashi identities for colour-ordered correlation functions associated to Yangian symmetries in integrable planar gauge theories. Gauge fixing represented a major complication in formulating these relations and their underlying Slavnov–Taylor identities. In order to account for the extraneous ghost fields and gauge

degrees of freedom, the identities possess a number of auxiliary terms to cancel off contributions from these unphysical degrees of freedom. An exciting side-effect of these terms is that they involve novel types of non-local gauge and BRST symmetries which extend to arbitrary non-integrable planar gauge theories and their gauge-fixed counterparts and may serve some purpose there. Our derivation of Slavnov–Taylor identities relied on some conjectural bi-local total variations of the *planar* path integral involving some form of bi-local quantum correlators. Fortunately, the resulting Ward–Takahashi identities can be formulated using conventional notions, and we demonstrated that they do indeed hold true for several tree-level correlation functions with up to four external legs as well as the one-loop correlator of three fields. Importantly, these identities, tools and verifications crucially rely on the planar limit. This further justifies the relevance of the planar limit for integrability and Yangian symmetry to apply for the considered models.

The main open question related to our work is whether the Slavnov–Taylor identities continue to hold for more legs and in the quantum theory, either perturbatively or non-perturbatively. Given the successes in applying integrability to the determination of various observables at higher loops or even at finite coupling strength, it is hardly conceivable that they will ever fail. Nevertheless, it remains to be proven that the structure of Feynman diagrams at loop level is compatible with the Slavnov–Taylor identities. Will the symmetry generators require correction terms in the renormalised theory? Or will divergences associated to the unphysical degrees of freedom violate the symmetry? Furthermore, there is the logical possibility that Yangian symmetry is anomalous in the quantum theory; this option remains to be rigorously excluded. Nevertheless, we have sketched an argument to achieve the latter for the main example of planar $\mathcal{N} = 4$ sYM, but it is based on several assumptions which remain to be substantiated. Altogether, we believe that this work has laid the foundations for addressing and understanding Yangian symmetry in the quantum theory.

Correlation functions are not just useful in their own right, but they can also be used to formulate other relevant observables such as scattering amplitudes (via the LSZ reduction) and Wilson loop expectation values (by integrating over the insertions points). The Slavnov–Taylor or Ward–Takahashi identities can then be translated to corresponding symmetry statements for these observables. For Wilson loop expectation values one should be able to make contact to the results of [23, 34] more or less directly. For scattering amplitudes, this should open up a path to rigorously derive the Yangian-based differential equations of [20]. Similarly, the Ward–Takahashi identities can be expected to explain the non-linear representation on scattering amplitudes with collinear particle momenta [35]. Along the same lines, it may be interesting to understand whether the identities continue to hold for correlation functions with a singular configuration of points such as light-like separated external fields. The case of coincident fields is directly linked to the notion of composite local operators, and the representation of Yangian symmetry on them [2] should also follow from our framework. Moreover, it would be interesting to understand violations of Yangian symmetry related to certain observables. For example, corners and cusps in Wilson loops generate divergences which can render Ward identities anomalous [17].

This work also raised some technical questions for planar quantum field theories. It would be highly desirable to understand better the bi-local variational identity (5.5) for the planar path integral as well as the notion of bi-local quantum correlators $\langle \mathcal{O}_1 \wedge \mathcal{O}_2 \rangle$. With a proper definition of the latter and a proof of the former, all the transformations that led to the Slavnov–Taylor identities could be carried out rigorously; this would establish Ward–Takahashi identities for arbitrarily many external legs and at loop level (up to regularisation, renormalisation and anomalies). Another curious observation pointed out at the end of Sec. 6.3 is that the Yangian representations on the action and on correlation functions show some structural differences w.r.t. overlapping contributions.

Finally, (Yangian) quantum algebras applied to planar gauge theory and non-linear representations deserve a deeper mathematical understanding. How to complete the Yangian relations which are intertwined with gauge transformations and non-local generalisations thereof? How to formulate the Serre-relations precisely in this case? How can the non-linear action on cyclic states be interpreted in terms of representation theory of the algebra? How to interpret the overlapping terms and how do they relate to the coalgebra structure? It would also be interesting to understand whether there is any relationship to the non-ultra-local issues of the corresponding symmetry algebra for the non-linear sigma model for the string theory worldsheet theory, see [36, 37]. Last but not least, one may wonder whether there is there a relationship to the anomaly considerations for the worldsheet theory in [38]?

## Acknowledgements

We would like to thank Matteo Rosso for his collaboration at earlier stages of the present work. We also thank Florian Loebbert and Cornelius Schmidt-Colinet for discussions of aspects related to this article.

The work of NB and AG is partially supported by grant no. 615203 from the European Research Council under the FP7 and by the Swiss National Science Foundation through the NCCR SwissMAP. This research was supported in part by the U.S. National Science Foundation under Grant No. NSF PHY-1125915.

## A  Ward–Takahashi identity with gauge-fixing terms

In this appendix we verify some Ward–Takahashi identities with all gauge-fixing terms made explicit. The transformations will be performed in terms of diagrams as explained in Sec. 6.2.

### A.1  Propagators

We start by summarising the symmetry properties of propagators, see Sec. 6.1, in a diagrammatical notation. The defining property of propagators (6.18) reads

$$
\text{—}\bigcirc\text{—} = i \,\text{————}. \tag{A.1}
$$

Invariance of the propagator under BRST transformations takes the form (6.1)

$$
0 = \text{—}\triangleright\text{—} = i\,\triangleright\text{————} + i\,\text{————}\triangleleft\,. \tag{A.2}
$$

In these diagrams the (red) triangle represents the BRST generator Q. This will be used to tacitly shift BRST generators from one end of a propagator to the other

$$
\triangleright\text{————} = -\,\text{————}\triangleleft\,. \tag{A.3}
$$

Invariance of the propagator under level-zero transformations receives some extra terms compared to (6.23) due to gauge fixing (6.4)

$$
\begin{aligned}
0 = \text{—}\ominus\text{——} &= \text{—}\bigcirc\!\triangleright\text{—} + \text{—}\triangleleft\!\bigcirc\text{—} - \text{—}\triangleleft\boxed{\ }\text{——} - \text{——}\boxed{\ }\triangleright\text{—} \\
&= i\,\triangleright\text{————} + i\,\text{————}\triangleleft + \triangleright\boxed{\ }\text{——} + \text{——}\boxed{\ }\triangleleft\,.
\end{aligned} \tag{A.4}
$$

Here, the (purple) rectangle represents the BRST compensator $\mathscr{K}[J]$ associated to the enclosed generator J. The latter two terms require special attention due to exchange statistics

of the BRST generator Q and the compensator $\mathscr{K}[J]$ which cannot be expressed well in the diagrams alone: Here we assume that the sign of a diagram is determined relative to the sequence $\ldots\mathscr{K}[J]\ldots Q\ldots$ in the corresponding mathematical expression. If an expression has the opposite ordering $\ldots Q\ldots\mathscr{K}[J]\ldots$, its diagram will receive an extra sign. This is in fact the reason for the negative sign for the terms in the first line which originate from the combination $+Q\cdot\mathscr{K}[J]$. Their sign in the second line is flipped due to (A.3).

## A.2 Local symmetries of three-point functions

Here we discuss the Ward–Takahashi identities corresponding to local symmetries for three external fields (6.13) using diagrams. The external fields will be summarised in the operator $\mathscr{O} := \mathrm{tr}(Z_1 Z_2 Z_3)$. We will work modulo cyclic permutations of the external fields to collapse several equivalent terms into one by means of the equivalence relation '$\simeq$'.

**BRST symmetry.** We first consider the identity for BRST symmetry. As BRST symmetry itself does not require compensators, the consideration is equivalent to Sec. 6.2 without gauge fixing where J is replaced by Q. In short, invariance of the three-point function follows using invariance of the propagator (6.1) as

$$\left\langle Q\cdot\mathscr{O}\right\rangle \simeq 3i \, \bigcirc + 3 \, \bigcirc = -3i \, \bigcirc - 3i \, \bigcirc \tag{A.5}$$

$$\simeq -i \, \bigcirc = -i\left\langle\tfrac{1}{3}(Q\cdot\mathscr{S})_{[3]}\,\mathscr{O}\right\rangle = 0.$$

**Level-zero symmetry.** Next we discuss the level-zero Ward–Takahashi identity for three external fields (6.13) using diagrams. It consists of the following terms (for convenience, we identify the diagrams with labels a–k)

$$\left\langle J_{[0]}\cdot\mathscr{O}\right\rangle_{[1]} \simeq 3i \, \bigcirc^{a},$$

$$\left\langle J_{[1]}\cdot\mathscr{O}\right\rangle_{[0]} \simeq 3 \, \bigcirc^{b},$$

$$-i\left\langle\tfrac{1}{3}\mathscr{K}[J]_{[3]}\,Q_{[0]}\cdot\mathscr{O}\right\rangle_{[0]} \simeq 3i \, \bigcirc^{c}, \tag{A.6}$$

$$-i\left\langle\tfrac{1}{2}\mathscr{K}[J]_{[2]}\,Q_{[1]}\cdot\mathscr{O}\right\rangle_{[0]} \simeq -3i \, \bigcirc^{d} - 3i \, \bigcirc^{e},$$

$$-i\left\langle\tfrac{1}{2}\mathscr{K}[J]_{[2]}\,Q_{[0]}\cdot\mathscr{O}\right\rangle_{[1]} \simeq 3 \, \bigcirc^{f} - 3 \, \bigcirc^{g} - 3 \, \bigcirc^{h}.$$

In diagrams c, g, h we have used the linearised BRST symmetry of propagators (A.3) to shift the BRST generator to the central vertex.

We can transform diagrams a and f by adding the linearised level-zero invariance of the gauge-fixed propagator (A.4),

$$0 = -3 \bigcirc \simeq -3i \bigcirc{}^{a} - 3 \bigcirc{}^{f} - 3i \bigcirc{}^{i} - 3 \bigcirc{}^{j} . \tag{A.7}$$

This effectively replaces them by diagrams i and j with the opposite sign. Then we add the BRST invariance of the action in (A.5) together with insertion of a spectator vertex $\mathcal{K}[J]_{[2]}$ to cancel most of the terms related to gauge fixing,

$$0 = 3 \bigcirc = \left\langle \tfrac{1}{2}\mathcal{K}[J]_{[2]} \tfrac{1}{3}(Q{\cdot}\mathcal{S})_{[3]} \mathcal{O} \right\rangle$$

$$\simeq 3 \bigcirc{}^{j} + 3 \bigcirc{}^{g} + 3 \bigcirc{}^{h} \tag{A.8}$$

$$+ 3i \bigcirc{}^{k} + 3i \bigcirc{}^{d} + 3i \bigcirc{}^{e} .$$

For diagrams d, e and k we have removed a quadratic vertex by means of (A.1). The remaining 4 terms represent the level-zero invariance of the action at three fields with gauge fixing. We add the corresponding transformation of the action,

$$0 = i \bigcirc = i\left\langle \tfrac{1}{3}(J{\cdot}\mathcal{S} + Q{\cdot}\mathcal{K}[J])_{[3]} \mathcal{O} \right\rangle$$

$$\simeq 3i \bigcirc{}^{i} - 3 \bigcirc{}^{b} - 3i \bigcirc{}^{c} - 3i \bigcirc{}^{k} . \tag{A.9}$$

For diagram b we have again removed a quadratic vertex. Note that the ordering of $\mathcal{K}[J]$ and Q in $Q{\cdot}\mathcal{K}[J]$ is opposite to the ordering in all the above expressions, hence the corresponding diagrams receive an extra sign flip. When adding up all the diagrams, we finally get zero

$$\left\langle J{\cdot}\mathcal{O} \right\rangle - i\left\langle \mathcal{K}[J] Q{\cdot}\mathcal{O} \right\rangle = 0. \tag{A.10}$$

### A.3 Bi-local symmetries of three-point functions

In the following we will discuss the Ward–Takahashi identities corresponding to bi-local symmetries for three external fields $\mathcal{O} := \mathrm{tr}(Z_1 Z_2 Z_3)$. Again, we will work modulo cyclic permutations of the external fields by means of the equivalence relation '$\simeq$'.

**Bi-local BRST symmetry.**  We start with the bi-local BRST generator $Q \otimes Q$ introduced in Sec. 4.3. Since the BRST component operators $Q$ have no compensators $\mathscr{K}[Q]$, the derivation is identical to the one presented in Sec. 6.3 together with a proper treatment of the gauge-fixed local contributions $\mathscr{K}[Q \otimes Q]$ along the lines of App. A.2. The Ward–Takahashi identity for $Q \otimes Q$ then reads

$$\big\langle (Q \otimes Q) \cdot \mathcal{O} \big\rangle - i \big\langle \mathscr{K}[Q \otimes Q] Q \cdot \mathcal{O} \big\rangle = 0. \tag{A.11}$$

The relevant invariance of the action takes the form

$$0 = \ \bigcirc \ = \Big\langle \tfrac{1}{3} \big( (Q \otimes Q) \cdot \mathscr{S} + Q \cdot \mathscr{K}[Q \otimes Q] \big)_{[3]} \mathcal{O} \Big\rangle$$

$$\tag{A.12}$$

$$\simeq 3i \ \bigcirc \ - \ \bigcirc \ - i \ \bigcirc \ + i \ \bigcirc \ - 3 \ \bigcirc \ .$$

**Mixed bi-local symmetry.**  Now we will verify the mixed symmetry $Q \wedge J$ introduced in Sec. 4.3 including the gauge-fixing corrections caught by the Slavnov–Taylor identity (5.9)

$$\big\langle (Q \wedge J) \cdot \mathcal{O} + i \mathscr{K}[J] (Q \otimes Q) \cdot \mathcal{O} \big\rangle$$
$$+ \big\langle \big( -i \mathscr{K}[Q \wedge J] + \mathscr{K}[J] \mathscr{K}[Q \otimes Q] \big) Q \cdot \mathcal{O} - i \mathscr{K}[Q \otimes Q] J \cdot \mathcal{O} \big\rangle = 0. \tag{A.13}$$

This Ward–Takahashi identity consists of five different contributions totalling 22 diagrams. In order to streamline the presentation, we will tacitly remove quadratic vertices by means of (A.1) and shift BRST generators $Q$ towards the centre of the diagram by means of (A.3). The diagrams turn out to be

$$\big\langle (Q \wedge J) \cdot \mathcal{O} \big\rangle \simeq 3 \ \bigcirc^{a} \ + i \ \bigcirc^{b_1} \ - i \ \bigcirc^{c_1}$$

$$+ \ \bigcirc^{d_1} \ - \ \bigcirc^{e_1} \ + \ \bigcirc^{f} \ - \ \bigcirc^{g} \ ,$$

$$i \big\langle \mathscr{K}[J] (Q \otimes Q) \cdot \mathcal{O} \big\rangle \simeq 3i \ \bigcirc^{h} \ + 3i \ \bigcirc^{i} \ + i \ \bigcirc^{j}$$

$$- \ \bigcirc^{b_2} \ + \ \bigcirc^{c_2} \ - \ \bigcirc^{k}$$

$$+ i \ \bigcirc^{d_2} \ - i \ \bigcirc^{e_2} \ - i \ \bigcirc^{l} \ + i \ \bigcirc^{m} \ ,$$

$$-i \big\langle \mathscr{K}[Q \wedge J] Q \cdot \mathcal{O} \big\rangle \simeq 3i \ \bigcirc^{n} \ ,$$

$$\langle \mathscr{K}[J]\mathscr{K}[Q\otimes Q]Q\cdot\mathscr{O}\rangle \simeq -3 \quad\text{(o)}\quad -3 \quad\text{(p)}\quad +3 \quad\text{(q}_2\text{)} ,$$

$$-i\langle \mathscr{K}[Q\otimes Q]J\cdot\mathscr{O}\rangle \simeq -3i \quad\text{(q}_1\text{)} . \tag{A.14}$$

Some remarks are in order: For the first set of diagrams with the regular application of the generator $Q \wedge J$ (as opposed to the earlier cases of $\widehat{J} = J^{(1)} \otimes J^{(2)}$ and $Q \otimes Q$) we note that the two underlying generators, $Q$ and $J$, are of different types and we have to explicitly consider both orderings.

Another subtlety concerns the signs due to statistics of the operators. Diagrams of the second term involve several fermionic operators $Q$ and terms $\mathscr{K}[J]$ whose ordering matters. We assume the reference ordering within mathematical expressions to be $(\mathscr{K}[J], Q, \dot{Q})$ where $\dot{Q}$ corresponds to the decorated BRST operator (triangle with dot) in the diagrams but otherwise acts as an ordinary BRST operator $Q$. Likewise, the reference ordering for fermionic operators $Q$ and terms $\mathscr{K}[J]$, $\mathscr{K}[Q\otimes Q]$ within diagrams in the fourth term is assumed to be $(\mathscr{K}[J], \mathscr{K}[Q\otimes Q], Q)$.

Next we would like to shift level-zero generators $J$ acting on external fields towards the centre of the diagrams $b_1$, $c_1$, $d_1$, $e_1$ and $q_1$. To this end we add a couple of terms each of which is zero using the extended invariance relation (A.4) of the gauge-fixed propagator,

$$0 = - \quad\text{(b)}\quad \simeq -i \quad\text{(b}_1\text{)}\quad + \quad\text{(b}_2\text{)}\quad -i \quad\text{(b}_3\text{)}\quad + \quad\text{(b}_4\text{)} ,$$

$$0 = + \quad\text{(c)}\quad \simeq +i \quad\text{(c}_1\text{)}\quad - \quad\text{(c}_2\text{)}\quad +i \quad\text{(c}_3\text{)}\quad - \quad\text{(c}_4\text{)} ,$$

$$0 = i \quad\text{(d)}\quad \simeq - \quad\text{(d}_1\text{)}\quad -i \quad\text{(d}_2\text{)}\quad - \quad\text{(d}_3\text{)}\quad -i \quad\text{(d}_4\text{)} ,$$

$$0 = - \quad\text{(e)}\quad \simeq + \quad\text{(e}_1\text{)}\quad +i \quad\text{(e}_2\text{)}\quad + \quad\text{(e}_3\text{)}\quad +i \quad\text{(e}_4\text{)} ,$$

$$0 = 3 \quad\text{(q)}\quad \simeq +3i \quad\text{(q}_1\text{)}\quad -3 \quad\text{(q}_2\text{)}\quad +3i \quad\text{(q}_3\text{)}\quad -3 \quad\text{(q}_4\text{)} .$$

Effectively this replaces diagrams $b_k$, $c_k$, $d_k$, $e_k$ and $q_k$ with $k = 1, 2$ by the corresponding diagrams with $k = 3, 4$. Note that the signs of the diagrams with $k = 2, 4$ are superficially different from the underlying relation (A.4). This is because the fermionic operators and terms are ordered as $(Q, \mathscr{K}[J], \dot{Q})$ and $(\mathscr{K}[Q\otimes Q], \mathscr{K}[J], Q)$. Therefore they require one elementary permutation to be brought to the assumed ordering corresponding to a sign flip.

Furthermore, we add the following combination of terms to our set of diagrams which is

zero by means of the invariance of the gauge-fixed action under Q ⊗ Q (A.12)

$$0 = -3 \ \text{⬡} \ = -\left\langle \mathscr{K}[J] \tfrac{1}{3}\big((Q \otimes Q)\cdot \mathscr{S} + Q\cdot \mathscr{K}[Q \otimes Q]\big)_{[3]} \mathscr{O} \right\rangle$$

$$\simeq -3i \ \text{(h)} \ -3i \ \text{(i)} \ -3i \ \text{(r)}$$

$$- \ \text{(b}_4) \ + \ \text{(c}_4) \ + \ \text{(k)}$$

$$+ i \ \text{(d}_4) \ + i \ \text{(l)} \ + i \ \text{(s)}$$

$$- i \ \text{(e}_4) \ - i \ \text{(m)} \ - i \ \text{(t)}$$

$$+ 3 \ \text{(o)} \ + 3 \ \text{(p)} \ + 3 \ \text{(q}_4) \ .$$

(A.15)

Note that we need to explicitly average over all cyclic permutation in (A.12) due to the extra vertex $\mathscr{K}[J]$ which breaks this symmetry.

Finally, we add the invariance of the gauge-fixed action under (4.25)

$$0 = i \ \text{⬡} \ = i\left\langle \tfrac{1}{3}\big((Q \wedge J)\cdot \mathscr{S} + (Q \otimes Q)\cdot \mathscr{K}[J] - J\cdot \mathscr{K}[Q \otimes Q] - Q\cdot \mathscr{K}[Q \wedge J]\big)_{[3]} \mathscr{O} \right\rangle,$$

(A.16)

with the four contributions

$$i\left\langle \tfrac{1}{3}\big((Q \wedge J)\cdot \mathscr{S}\big)_{[3]} \mathscr{O} \right\rangle \simeq -3 \ \text{(a)} \ + i \ \text{(b}_3) \ - i \ \text{(c}_3)$$

$$+ \ \text{(d}_3) \ - \ \text{(e}_3) \ - \ \text{(f)} \ + \ \text{(g)} \ ,$$

$$-i\left\langle \tfrac{1}{3}(Q\cdot \mathscr{K}[Q \wedge J])_{[3]} \mathscr{O} \right\rangle \simeq -3i \ \text{(n)} \ ,$$

$$-i\left\langle \tfrac{1}{3}(J\cdot \mathscr{K}[Q \otimes Q])_{[3]} \mathscr{O} \right\rangle \simeq -3i \ \text{(q}_3) \ ,$$

$$i\left\langle \tfrac{1}{3}((\mathrm{Q}\otimes\mathrm{Q})\cdot\mathscr{K}[\mathrm{J}])_{[3]}\,\mathscr{O}\right\rangle \simeq 3i\,\bigcirc - i\,\bigcirc - i\,\bigcirc + i\,\bigcirc .$$

(A.17)

Altogether we find that all terms cancel. More explicitly, every one of the 35 distinct diagrams is labelled by a letter, and it appears twice with equal but opposite coefficients. This shows that the Ward–Takahashi identity (A.13) indeed holds in this case.

Even more, all the intermediate terms for the Slavnov–Taylor identity in the last few lines of (5.8) are produced with the expected coefficients. Among these, the bi-local correlator

$$-i\left\langle\big(\mathrm{J}\cdot\mathscr{S}+\mathrm{Q}\cdot\mathscr{K}[\mathrm{J}]\big)\wedge(\mathrm{Q}\cdot\mathscr{O})\right\rangle,$$

(A.18)

for which we have merely provided a superficial description in Sec. 5.2, is apparently represented truthfully by diagrams b, c, d, e in (A.14). All of this gives us some confidence that the considerations in Sec. 5 apply indeed, and that we can trust the Slavnov–Taylor identities for bi-local symmetries.

**Yangian symmetry.** We have also sketched the corresponding calculation for invariance of the gauge-fixed three-point correlator under a level-one Yangian generator. Unfortunately, it involves a substantial increase in combinatorics compared to the previous calculations due to the various types of elements that contribute. For example, the Ward–Takahashi identity (5.10) expands to 54 diagrammatical terms. An initial analysis shows that all kinds of diagrams may indeed cancel by using invariances of the action. However, a more careful investigation would be needed to convincingly demonstrate that the identity holds for three external fields at tree level. For example, this would be desirable to confirm the combination of terms in (5.10) including its combinatorial factors of $^{1}/_{2}$.

# B  Level-one invariance of the three-point function at one loop

In this appendix we show the Yangian invariance of the three-point function at one loop explicitly (modulo gauge fixing and regularisation). Recall that symmetry variation of the correlator yields

$$\left\langle\widehat{\mathrm{J}}'\cdot(Z_1 Z_2 Z_3)\right\rangle_{(1)}$$

$$\simeq -i\,\bigcirc - 3\,\bigcirc - \bigcirc + \bigcirc + i\,\bigcirc$$

$$- i\,\bigcirc - i\,\bigcirc + i\,\bigcirc - 3\,\bigcirc - \bigcirc + \bigcirc$$

$$- \bigcirc - \bigcirc + \bigcirc + 3i\,\bigcirc + i\,\bigcirc - i\,\bigcirc$$

$$- 3\,\bigcirc - \bigcirc - \bigcirc + i\,\bigcirc$$

$$- 3\,\bigcirc + \bigcirc + \bigcirc - i\,\bigcirc .$$

(B.1)

We can cancel all diagrams by adding the following terms, all of which are zero by invariance of the action and commutativity of the level-zero generators

$$-3i\ \simeq +i\ -i\ +i\ +3\ -\ +$$
(B.2)
$$+3\ -\ +\ +3\ +\ -\ ,$$

$$+\frac{i}{2}\ \simeq +\frac{i}{2}\ -\frac{i}{2}\ +\frac{i}{2}\ +\frac{1}{2}\ +\frac{1}{2}\ +\frac{1}{2}\ ,$$
(B.3)

$$-\frac{i}{2}\ \simeq +\frac{i}{2}\ +\frac{i}{2}\ +\frac{i}{2}\ -\frac{1}{2}\ -\frac{1}{2}\ -\frac{1}{2}\ ,$$
(B.4)

$$-i\ \simeq -i\ -2i\ -\ -\ +\ ,$$
(B.5)

$$+\frac{i}{2}\ \simeq +\frac{i}{2}\ +\frac{i}{2}\ +\frac{1}{2}\ -\frac{1}{2}\ +\frac{1}{2}\ ,$$
(B.6)

$$-\frac{i}{2}\ \simeq -\frac{i}{2}\ +\frac{i}{2}\ -\frac{1}{2}\ +\frac{1}{2}\ -\frac{1}{2}\ ,$$
(B.7)

$$-\frac{1}{2}\ \simeq +\frac{1}{2}\ +\frac{1}{2}\ +\frac{1}{2}\ -\frac{i}{2}\ -\frac{i}{2}\ -\frac{i}{2}\ ,$$
(B.8)

$$+\frac{1}{2}\ \simeq -\frac{1}{2}\ -\frac{1}{2}\ +\frac{1}{2}\ -\frac{i}{2}\ +\frac{i}{2}\ +\frac{i}{2}\ ,$$
(B.9)

$$-3i\ \simeq +i\ -i\ +i\ +3\ -\ +\ ,$$ (B.10)

$$-3i\ \simeq -i\ -i\ +i\ +3\ +\ -\ ,$$ (B.11)

$$-3i\ \simeq +i\ +i\ -i\ +3\ -\ +$$
(B.12)
$$+3\ -\ +\ +3\ +\ -\ ,$$

$$+i\ \simeq +i\ +i\ +\ ,$$
(B.13)

$$-i\;\text{[diagram]} \simeq -i\;\text{[diagram]} + i\;\text{[diagram]} - \text{[diagram]}\,, \tag{B.14}$$

$$+i\;\text{[diagram]} \simeq +i\;\text{[diagram]} - i\;\text{[diagram]} - \text{[diagram]}\,, \tag{B.15}$$

$$-i\;\text{[diagram]} \simeq -i\;\text{[diagram]} - i\;\text{[diagram]} + \text{[diagram]}\,, \tag{B.16}$$

$$+i\;\text{[diagram]} \simeq +i\;\text{[diagram]} - i\;\text{[diagram]} - i\;\text{[diagram]} + \text{[diagram]}\,, \tag{B.17}$$

$$-i\;\text{[diagram]} \simeq -i\;\text{[diagram]} + i\;\text{[diagram]} + i\;\text{[diagram]} - \text{[diagram]}\,, \tag{B.18}$$

$$+i\;\text{[diagram]} \simeq -i\;\text{[diagram]} + i\;\text{[diagram]} + i\;\text{[diagram]} + \text{[diagram]}\,, \tag{B.19}$$

$$-i\;\text{[diagram]} \simeq +i\;\text{[diagram]} - i\;\text{[diagram]} - i\;\text{[diagram]} - \text{[diagram]}\,, \tag{B.20}$$

$$-3\;\text{[diagram]} \simeq +\text{[diagram]} - \text{[diagram]} + \text{[diagram]} - 3i\;\text{[diagram]} + i\;\text{[diagram]} - i\;\text{[diagram]}\,, \tag{B.21}$$

$$
\begin{aligned}
-3\;\text{[diagram]} \simeq\; &+\frac{3}{2}\text{[diagram]} - \frac{3}{2}\text{[diagram]} + \frac{3}{2}\text{[diagram]} - \frac{3}{2}\text{[diagram]} \\
&-3\;\text{[diagram]} + \frac{3}{4}\text{[diagram]} - \frac{3}{4}\text{[diagram]} + \frac{3}{4}\text{[diagram]} - \frac{3}{4}\text{[diagram]} \\
&-3\;\text{[diagram]} - \frac{3}{4}\text{[diagram]} + \frac{3}{4}\text{[diagram]} - \frac{3}{4}\text{[diagram]} + \frac{3}{4}\text{[diagram]} \\
&-3\;\text{[diagram]} - \frac{3}{4}\text{[diagram]} + \frac{3}{4}\text{[diagram]} - \frac{3}{4}\text{[diagram]} + \frac{3}{4}\text{[diagram]} \\
&-3\;\text{[diagram]} + \frac{3}{4}\text{[diagram]} - \frac{3}{4}\text{[diagram]} + \frac{3}{4}\text{[diagram]} - \frac{3}{4}\text{[diagram]} \\
&+\frac{3i}{4}\text{[diagram]} - \frac{3i}{4}\text{[diagram]} - \frac{3i}{4}\text{[diagram]} + \frac{3i}{4}\text{[diagram]}\,,
\end{aligned}
\tag{B.22}
$$

$$+\text{[diagram]} \simeq +\text{[diagram]} + \text{[diagram]} - i\;\text{[diagram]}\,, \tag{B.23}$$

$$-\text{[diagram]} \simeq -\text{[diagram]} + \text{[diagram]} + i\;\text{[diagram]}\,, \tag{B.24}$$

$$+ \;\; \text{(diagram)} \;\; \simeq \; + \;\; \text{(diagram)} \; - \;\; \text{(diagram)} \; - \;\; \text{(diagram)} \; - \; i \;\; \text{(diagram)} \;,$$

(B.25)

$$- \;\; \text{(diagram)} \;\; \simeq \; - \;\; \text{(diagram)} \; + \;\; \text{(diagram)} \; + \;\; \text{(diagram)} \; + \; i \;\; \text{(diagram)} \;,$$

(B.26)

$$+ \frac{3}{4} \;\; \text{(diagram)} \;\; \simeq \; - \frac{3}{4} \;\; \text{(diagram)} \; - \frac{3}{4} \;\; \text{(diagram)} \; - \frac{3}{4} \;\; \text{(diagram)}$$
$$+ \frac{3}{4} \;\; \text{(diagram)} \; - \frac{3}{4} \;\; \text{(diagram)} \; + \frac{3}{4} \;\; \text{(diagram)} \; - \frac{3}{4} \;\; \text{(diagram)} \;,$$

(B.27)

$$- \frac{3}{4} \;\; \text{(diagram)} \;\; \simeq \; - \frac{3}{4} \;\; \text{(diagram)} \; + \frac{3}{4} \;\; \text{(diagram)} \; + \frac{3}{4} \;\; \text{(diagram)}$$
$$+ \frac{3}{4} \;\; \text{(diagram)} \; - \frac{3}{4} \;\; \text{(diagram)} \; - \frac{3}{4} \;\; \text{(diagram)} \; + \frac{3}{4} \;\; \text{(diagram)} \;,$$

(B.28)

$$+ \frac{1}{4} \;\; \text{(diagram)} \;\; \simeq \; - \frac{1}{4} \;\; \text{(diagram)} \; + \frac{1}{4} \;\; \text{(diagram)} \; + \frac{1}{4} \;\; \text{(diagram)}$$
$$+ \frac{1}{4} \;\; \text{(diagram)} \; - \frac{1}{4} \;\; \text{(diagram)} \; - \frac{1}{4} \;\; \text{(diagram)} \; + \frac{1}{4} \;\; \text{(diagram)} \;,$$

(B.29)

$$- \frac{1}{4} \;\; \text{(diagram)} \;\; \simeq \; - \frac{1}{4} \;\; \text{(diagram)} \; - \frac{1}{4} \;\; \text{(diagram)} \; - \frac{1}{4} \;\; \text{(diagram)}$$
$$+ \frac{1}{4} \;\; \text{(diagram)} \; - \frac{1}{4} \;\; \text{(diagram)} \; + \frac{1}{4} \;\; \text{(diagram)} \; - \frac{1}{4} \;\; \text{(diagram)} \;,$$

(B.30)

$$- \frac{1}{2} \;\; \text{(diagram)} \;\; \simeq \; + \frac{1}{2} \;\; \text{(diagram)} \; - \frac{1}{2} \;\; \text{(diagram)} \; + \frac{1}{2} \;\; \text{(diagram)} \; + \frac{i}{2} \;\; \text{(diagram)} \;,$$

(B.31)

$$- \frac{1}{2} \;\; \text{(diagram)} \;\; \simeq \; + \frac{1}{2} \;\; \text{(diagram)} \; - \frac{1}{2} \;\; \text{(diagram)} \; - \frac{1}{2} \;\; \text{(diagram)} \;,$$

(B.32)

$$+ \frac{1}{2} \;\; \text{(diagram)} \;\; \simeq \; - \frac{1}{2} \;\; \text{(diagram)} \; + \frac{1}{2} \;\; \text{(diagram)} \; - \frac{1}{2} \;\; \text{(diagram)} \; - \frac{i}{2} \;\; \text{(diagram)} \;,$$

(B.33)

$$+ \frac{1}{2} \;\; \text{(diagram)} \;\; \simeq \; - \frac{1}{2} \;\; \text{(diagram)} \; + \frac{1}{2} \;\; \text{(diagram)} \; + \frac{1}{2} \;\; \text{(diagram)} \;,$$

(B.34)

$$- \;\; \text{(diagram)} \;\; \simeq \; + \;\; \text{(diagram)} \; - \;\; \text{(diagram)} \; - \;\; \text{(diagram)} \; - \; i \;\; \text{(diagram)} \;,$$

(B.35)

$$- \;\; \text{(diagram)} \;\; \simeq \; + \;\; \text{(diagram)} \; + \;\; \text{(diagram)} \; + \;\; \text{(diagram)} \; - \; i \;\; \text{(diagram)} \;,$$

(B.36)

$$+ \;\;\text{(diagram)} \;\simeq\; - \;\text{(diagram)} \;+\; \text{(diagram)} \;+\; \text{(diagram)} \;+\; i \;\text{(diagram)} \;,\tag{B.37}$$

$$+ \;\;\text{(diagram)} \;\simeq\; - \;\text{(diagram)} \;-\; \text{(diagram)} \;-\; \text{(diagram)} \;+\; i \;\text{(diagram)} \;,\tag{B.38}$$

$$+ \tfrac{i}{8} \;\text{(diagram)} \;\simeq\; + \tfrac{i}{4} \;\text{(diagram)} \;+\; \tfrac{i}{4} \;\text{(diagram)} \;,\tag{B.39}$$

$$- \tfrac{i}{8} \;\text{(diagram)} \;\simeq\; - \tfrac{i}{4} \;\text{(diagram)} \;-\; \tfrac{i}{4} \;\text{(diagram)} \;.\tag{B.40}$$

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
