# Peer review of "Yangian Algebra and Correlation Functions in Planar Gauge Theories"

_SciPost Physics, doi:SciPost Phys. 5, 018 (2018)_

## Round 1 · Referee Report · Anonymous · 2018-6-22

Strengths

1-Significant contribution to the understanding of integrability in planar N=4 SYM
2-New Ward-Takahashi identities associated to an infinite-dimensional symmetry algebra

Weaknesses

1-The paper is quite technical

Report

Despite the very many successes of quantum integrability applied to planar N=4 SYM, there is still a substantial gap between the powerful, but sometimes ad-hoc, computational methods and the first principle definition of the theory. It is therefore very important to try to close this gap and develop a rigorous framework from which the integrability techniques arise. The paper ``Yangian algebra and correlation functions in planar gauge theories'' by Niklas Beisert and Aleksander Garus provides a significant step towards this goal. In this paper, the authors continue their effort to understand from first principles the Yangian invariance, which is the foundation for integrability, in the context of planar gauge theories. It extends their previous work, where they established Yangian invariance of the action for such theories, and supplements it with implications of Yangian symmetry for correlation functions of local operators. In particular, the authors propose a novel set of identities which play the role of the Ward-Takahashi identities associated to Yangian symmetry.

Many claims presented in the paper are still conjectural and further tests are necessary to fully confirm the conjectured identities. However, the authors provide substantial evidence that their assertions are correct and they carefully check that the Yangian symmetry can be formulated at the level of correlation functions.

The paper is rather technical in nature, however, it contains the proper amount of details and an extensive discussion of limitations of the method. The obtained results are original and interesting and they provide a significant addition to our understanding of integrability in N=4 SYM, and to the study of planar gauge theories in general.

Requested changes

1- The use of notation is confusing at times, if not incorrect in places. In particular, the authors use both $J^{(1)}$ and $J^1$ to refer to the same algebra generator, as far as I can say. I suggest the authors should improve on that.
2-(optional) There is no comment in the paper on higher-level generators of the Yangian algebra. Although they can be obtained from level-zero and level-one generators, it is important, from the integrability point of view, that the Yangian algebra is infinite dimensional. It poses a natural question on how the construction proposed by the authors extends also to higher-order generators: are the bi-local gauge symmetries sufficient to generate all gauge symmetries associated to higher-level generators or do they need to be extended further?

  • validity: high
  • significance: high
  • originality: top
  • clarity: good
  • formatting: excellent
  • grammar: excellent

Author:  Niklas Beisert  on 2018-07-02  [id 285]

(in reply to Report 1 on 2018-06-22)
Category:
remark
answer to question
suggestion for further work

We thank the referee for the useful feedback on the manuscript.

Concerning the requested changes:

  1. The notation J^1,2 for some generic local/level-zero generator as opposed to J^(1,2) for Sweedler's notation of a part of Jhat was not explained carefully. Thanks for pointing this out. Nevertheless we hope that we used the two distinct notations consistently. To make the distinction clearer, we shall change J^1,2 to J^a,b where a,b label generic choices of level-zero generators.

  2. An infinite amount of (local?) conserved quantities (in involution) is certainly required for integrability, but the size of the Yangian algebra is a bit subtle: As a (deformed) universal enveloping algebra it is certainly infinite-dimensional. Also the underlying loop algebra is infinite-dimensional (this corresponds to the higher levels we omitted). However, the algebra is generated by finitely many elements (level-zero and level-one generators) and at the end of the day it is only these that give rise to independent symmetry relations. In that sense, integrabilty curiously depends on only finitely many symmetry relations to hold. We were never really sure what to make of these facts, but they equally apply to generic integrable models with Yangian and quantum affine algebras and are thus not specific to our paper. Nevertheless, we will add a brief note regarding the higher-level generators below (2.13).

Concerning the question in the second part of 2.: higher-level/multi-local gauge transformations certainly exist in this framework and under certain conditions (even the bi-local gauge transformations rely on some conditions to hold) they will be symmetries. Some of these will be generated by commutators of bi-local gauge transformations, but it is not evident whether all of them arise in this way. Some kind of Serre relations (which we largely excluded from our treatment) may also play a role there. It will be interesting to follow the construction of a gauge-extended Yangian algebra and its ideals, but fortunately it appears to be consistent to restrict to level-zero and level-one generators as everything relevant for our study follows from them. Therefore we'd rather not mention this issue in the present paper.

---

## Round 1 · Referee Report · Anonymous · 2018-6-24

Strengths

1. Provides a detailed account of the action of bi-local symmetries, for example level-one Yangian
generators, on cyclic products of adjoint fields. In particular the role of field dependent gauge
transformations in the algebra of symmetries is elucidated and leads to the introduction of a new class
of bi-local symmetries which are a combination of gauge transformations and symmetry generators.

2. Gives a careful treatment of gauge fixing which is a necessary prerequisite for understanding
the consequences of the symmetries for physical quantities such as correlation functions.

3. Gives explicit computations at tree-level and one-loop to verify Ward identities, which are based on
a conjectured bi-local variation, for level-one bi-local Yangian generators. A useful graphical method
is described to aid in these calculations.

Weaknesses

1. The paper does require the reader to already be quite familiar with the previous works on the topic, in
particular 1701.09162 and 1803.06310. While the discussion is quite general for the most part, at several
points specific properties of N=4 SYM are used which are not previously described in any detail.

2. The evidence for the existence of the bi-local symmetries at loop level is significantly weaker than at
tree-level.

Report

Integrability in the planar limit of N=4 SYM has proven to be immensely useful in the calculation of the
spectrum of anomalous dimensions but also for scattering amplitudes, Wilson loops and other
quantities. The underlying reason for this integrability is however less well understood. This work
proposes Ward identities for correlation functions of fields and checks their validity in a number of
examples at tree and one-loop level. This work thus provides an important step toward a more
fundamental and unified understanding of integrability in planar gauge theories.

There are a number of significant open issues regarding the proposed Yangian symmetry and the
corresponding Ward identities in this work. In particular it is not clear that the generators satisfy the
Serre relations and thus that the algebra is indeed that of the Yangian. Even more importantly for the
validity of the Ward identities is that the bi-local variation, while natural looking, is simply
conjectured rather than derived. The authors are however quite explicit about the status of both of
these issues and, at least for the latter at tree-level, the explicit checks carried out do provide
significant support for the conjecture.

Overall the paper is well written and laid out. While the paper describes a number of involved algebraic
calculations these are clearly described, with explicit examples given, and a useful diagrammatic
method is introduced which should be useful in future calculations.

Requested changes

1. As mentioned, the paper does require the reader to be familiar with previous work. A complete review
would be too much but a bit more detail in certain places could be helpful. Specifically, it is not
manifestly clear where and why the planar limit is needed and given how central this is perhaps it could
be stated explicitly. Perhaps the discussion near equation (2.1) where the need for the large-N limit is
introduced can be expanded slightly.

2. The authors are commendably clear regarding the conjectural status of their bi-local variation so
perhaps it would be better to say in the conclusion that the Ward identities are "proposed" rather than
"derived".

3. Finally a very minor point: the authors use "Gladly, ..." at several points which seems non-idiomatic
and "Fortunately, ..." or maybe "Happily, ..." might be improvements.

  • validity: top
  • significance: top
  • originality: top
  • clarity: top
  • formatting: perfect
  • grammar: excellent

Author:  Niklas Beisert  on 2018-07-02  [id 286]

(in reply to Report 2 on 2018-06-24)
Category:
remark
answer to question

We thank the referee for the useful feedback on the manuscript.

Unfortunately, we cannot quite follow the comment on weakness 1 because it is unspecific about the concrete issues that might be improved: Surely a complete understanding of the topic requires arXiv:1803.06310 (as well as many other papers on integrability, Yangian symmetry, N=4 sYM and AdS/CFT). However, we attempted to design this article to be as independent as possible by plainly mentioning the relevant features of the considered models without discussing or interpreting them (hoping that the interested reader would dig deeper). Therefore it would be interesting to understand which aspects we have missed to introduce to make our article easier to follow.

In fact, we hardly mention N=4 sYM at all. Particular features of N=4 sYM (beyond being an SU(N) gauge theory with adjoint matter) are mentioned in section 7.1 and 7.4. Also the relationship (2.49) is a special feature for a model to be potentially integrable. Beyond these, we did not identify any potentially relevant points and we do not know how to substantially improve weakness 1. Nevertheless we will make minor adjustments in these sections in order to introduce some terminology.

Concerning the requested changes:

  1. The role of the planar limit has always appeared to be somewhat elusive in this study. For the invariance of the action only the independence of field monomials seems to play a role. For this to work out, a rank N of at least 4 might actually do, but that is still very far away from the planar limit. This was already discussed at some length in arXiv:1803.06310. However, the Ward-Takahashi identities really seem to require a stronger version of the planar limit. In fact, the various identities we proposed hardly make any sense if not in the planar limit. We stated clearly that we expect them to be identities of the planar path integral. We will add some further clarification on the independence of monomials below (2.1) and also more strongly emphasise the role of the planar limit in the conclusions. There may of course be further aspects which we are not aware of yet.

  2. Good idea. We will change the wording in the beginning of the introduction as suggested. Nevertheless we hope it is fair to maintain the claim that we derived the Ward-Takahashi/Slavnov-Taylor identities from other conjectural variational identities of the planar path integral.

  3. Thanks, we will be glad to change the words.

---

## Round 2 · List of Changes

below (2.1) added:
"This means that in the planar limit N_c->infty all monomials are independent and form a basis for polynomials."

section 2.2 and following:
changed J^1,2 to J^a,b

above (2.12):
changed "Yangian algebra" to "Yangian quantum algebra"

below (2.13) added:
"Furthermore, the Yangian algebra contains infinitely many higher-level generators. However, we can ignore these because their algebraic relations are fully determined by the above relations involving only level-zero and level-one generators."

minor changes of wording:

first paragraph on page 13:
after "The above bi-local action (2.46) was derived" added "in [5]".
before "Let us comment on these conditions." added "The superconformal symmetries J^a and the action S of integrable planar gauge theories such as N=4 sYM satisfy these requirements."

added below (2.48):
Allowing for a_n,m != 0 will turn out unnecessary in our analysis.

expanded text around (2.49)

section 6.2, first sentence:
added "planar" to "correlators"

fixed the labels on one diagram

below (6.32), above (7.1):
added "planar" to "diagrams"

section 7.4 between (7.10) and (7.15):
various minor changes and additions to make arguments more self-contained

conclusions, first paragraph:
some changes of words. added remark on planar limit:
"Importantly, these identities, tools and verifications crucially rely on the planar limit. This further justifies the relevance of the planar limit for integrability and Yangian symmetry to apply for the considered models."

---

## Editorial Decision

published